

# A climate-dependent global model of ammonia emissions from chicken farming

Jize Jiang[1], David S. Stevenson[1], Aimable Uwizeye[2], Giuseppe Tempio[2] and Mark A. Sutton[3]

[1]School of GeoSciences, The University of Edinburgh, Crew Building, Alexander Crum Road, Edinburgh, EH9 3FF, UK
[2]Food and Agriculture Organization of the United Nations, Animal Production and Health Division, Viale delle Terme di Caracalla, 00153 Rome, Italy.
[3] UK Centre for Ecology and Hydrology, Edinburgh, Bush Estate, Midlothian, Penicuik, EH26 0QB, UK

*Correspondence to*: Jize Jiang (Jize.Jiang@ed.ac.uk)

## Abstract

Ammonia ($NH_3$) has significant impacts on the environment, which can influence climate and air quality, and cause acidification and eutrophication in terrestrial and aquatic ecosystems. Agricultural activities are the main sources of $NH_3$ emissions globally. Emissions of $NH_3$ from chicken farming are highly dependent on climate, affecting their environmental footprint and impact. In order to investigate the effects of meteorological factors and to quantify how climate change affect these emissions, a process-based model, AMmonia-CLIMate-Poultry (AMCLIM-Poultry) has been developed to simulate and predict temporal variations in $NH_3$ emissions from poultry excretion, here focusing on chicken farms and manure spreading. The model simulates the decomposition of uric acid to form total ammoniacal nitrogen which then partitions into gaseous $NH_3$ that is released to the atmosphere at hourly to daily resolution. Ammonia emissions are simulated by calculating nitrogen and moisture budgets within poultry excretion, including a dependence on environmental variables. By applying the model with global data for livestock, agricultural practice and meteorology, we calculate $NH_3$ emissions from chicken farming at global scale (0.5° resolution). Based on 2010 data, the AMCLIM-Poultry model estimates $NH_3$ emissions from global chicken farming of 5.5 Tg N yr$^{-1}$, about 13 % of the agriculture-derived $NH_3$ emissions. Taking account of partial control of the ambient environment for housed chicken (layers and broilers), the fraction of excreted nitrogen emitted as $NH_3$ is found to be up to three times larger in humid tropical locations than in cold or dry locations. For spreading of manure to land, rain becomes a critical driver affecting emissions in addition to temperature, with the emission fraction being up to five times larger in the semi-dry tropics than in cold, wet climates. The results highlight the importance of incorporating climate effects into global $NH_3$ emissions inventories for agricultural sources. The model shows increased emissions under warm and wet conditions, indicating that climate change will tend to increase $NH_3$ emissions over the coming century.

## 1 Introduction

Ammonia ($NH_3$) is the primary form of reactive nitrogen ($N_r$) which has significant impacts on the environment (Galloway et al., 2003; Sutton et al., 2013). Following its emission to the atmosphere, $NH_3$ readily reacts with gas phase acids to form particulate ammonium aerosols and may also condense onto existing particles (Fowler et al., 2009; Hertel et al., 2011). Gaseous $NH_3$ reacts with sulphuric acid ($H_2SO_4$) and nitric acid ($HNO_3$), which leads to formation of ammonium sulphate (($NH_4$)$_2SO_4$) and ammonium nitrate ($NH_4NO_3$) aerosols, respectively (Pinder et al., 2007, 2008; Hertel et al., 2011). These particles influence the radiation balance of the Earth by scattering light and altering the Earth's reflectivity (Xu and Penner, 2012), and also adversely affect regional air quality and human health (Brunekreef and Holgate, 2002; Pinder et al., 2007, 2008). The lifetime of atmospheric $NH_3$ is relatively short (hours to days) as it is removed rapidly by dry and wet deposition, or converted



to ammonium aerosols (Hendriks et al., 2016). Consequently, it is usually removed close to its source. In terrestrial ecosystems, acute exposure to $NH_3$ can cause visible foliar injury, reducing vegetation's tolerance to pests and diseases, especially for native plants and forests (Krupa 2003; Stulen et al., 1998; Sutton et al., 2011). Once deposited in water, $NH_3$ can result in acidification and eutrophication (Sutton et al., 2011). Excess $N_r$ input causes algal blooms in vulnerable aquatic ecosystems,

which harms local biodiversity.

The dominant source of $NH_3$ emission is from agricultural activities including animal housing, manure storage, and fertiliser usage for arable lands and crops. In western countries, approximately 80-90 % of atmospheric releases are from agriculture (Sutton et al., 2000; Hertel et al., 2011); a major source of $NH_3$ emission is from livestock waste. Oenema et al. (2007) estimated that $NH_3$ emissions cause a loss of approximately 19 % of nitrogen from livestock housing and manure storage, with a further

19 % being lost following the land application of manure. Previous studies that quantified $NH_3$ emissions from livestock have made estimations mainly by empirical methods. Emission factors were used, assuming fixed values for nitrogen volatilization rates, varying by animal type and management practices. For example, Misselbrook et al. (2000) derived $NH_3$ emission factors for major animals under various farming practices in UK agriculture. The advantage of this method is the relative simplicity for calculations. However, these emission factors only include climatic effects to a small extent. Using a fixed number to

describe the fraction of excreted nitrogen that volatilises as $NH_3$ does not always provide a realistic value under all environmental conditions and may cause large uncertainties in large scale estimations (e.g., when considering global scale estimates). Sommer and Hutchings (2001) reviewed a range of empirical models that were produced to predict $NH_3$ volatilization from slurry application to land. These models have experiment-derived equations. However, only one or two factors were studied and the interactions between these parameters were not investigated.

Another method for estimating $NH_3$ emission from livestock is to use process-based models based on a theoretical understanding of relevant processes, building on foundations developed for field sources (Sutton et al., 1995; Nemitz et al., 2001; Móring et al., 2016). Pinder at al. (2004) developed a process-based model for simulating $NH_3$ emissions from dairy cows, and the modelled $NH_3$ volatilization fraction from grazing, manure spreading and storage was shown to be reasonable compared to independent experimental data. Previous process modelling efforts for bird sources have focused on native seabird

populations (Riddick et al., 2016, 2018), using these as a natural laboratory to study the effect of global climate differences on $NH_3$ emissions, supported by a programme of measurements through different climates (Blackall et al., 2007; Riddick et al. 2015). Process-based models consider the effects of meteorological variation on the formation of $NH_3$ from an $N_r$ source, allowing calculation of $NH_3$ emissions that vary temporally and spatially. They can be extended to investigate the influences of various environmental conditions. However, as more complicated parameterizations are included in process-based models,

more detailed inputs are required, and lack of input data may limit the model's ability to obtain better results.

Ammonia emissions from animal waste are understood to be highly climate-sensitive. For example, Sutton et al. (2013) showed a factor of nine increase in emission rates between 5 °C and 25 °C, with additional effects from humidity and precipitation (Riddick et al., 2017).  Poultry numbers have increased roughly five-fold over the last 50 years (FAO, 2018), with chicken being the largest fraction. Global usage of poultry manure for land spreading increased from an estimated 5.0 Tg N yr$^{-1}$ in 2000

to 6.3 Tg N yr$^{-1}$ in 2010 (FAO, 2018). However, limited research has attempted to determine the magnitude of global $NH_3$ emissions from chicken farming whilst also considering climatic effects. In this study, a process-based model, AMmonia-CLIMate-Poultry (AMCLIM-Poultry) has been developed to simulate and predict temporal variations in $NH_3$ emissions from three major chicken production systems: (a) broilers, (b) layers and (c) backyard chicken, focusing on chicken housing and



land spreading of manure. The overarching goals of this study are to develop a process-based model and to apply it at global scale, to produce improved $NH_3$ emission estimates under influences of various meteorological factors, and to estimate total $NH_3$ emissions and their distribution for the present-day (year 2010) for chicken farming globally. Future work will quantify the estimated response of $NH_3$ emissions to climate change, the potential for year-to-year variability, and the implications for

$NH_3$ emissions from other livestock sectors.

## 2 Methods and Materials

### 2.1 Model description

Figure 1 shows activities related to chicken litter as a source of $NH_3$ emission in agricultural practices. Nitrogenous manure can be used as fertilisers on land or be stored for future usage. Typically, litter collected from chicken houses is spread on soils

for fertilising crops at the start of planting period, while excretion from backyard systems are applied fresh to fields or left on pastures and other ground. Ammonia can be released to the atmosphere through each of these activities. In this study, we developed the process-based AMCLIM-Poultry model to quantify $NH_3$ emissions from chicken farming, focusing on housing and manure land spreading. For this purpose, it is assumed in the model that emissions from stored manure occur within the animal house ('in-house storage') or do not behave significantly differently. The uncertainties associated with this

simplification are considered in Sect. 4.3.1.

The model has been developed from the GUANO model (Riddick et al., 2017), which simulated $NH_3$ emissions from wild seabird colonies. Major advances in the present study include the distinction between indoor and outdoor emissions, conservation of nitrogen between these stages, a new approach to simulate indoor emissions, and the linking of land spreading of chicken manure to the timing of agricultural cropping cycles. We used chicken excretion-derived nitrogen as an input

(described in Sect. 2.4.1), and incorporated meteorological factors to predict temporal variations of the $NH_3$ emissions. The simulations followed $N_r$ through the decomposition processes that uric acid (UA, solid/aqueous phase) in excretion hydrolyses to form total ammoniacal nitrogen (TAN = $NH_3$ + $NH_4^-$, aqueous phase), which then partitions to form gaseous $NH_3$ that is released to the atmosphere (Fig. 2). The quantitative equations used in the model are described below using SI units (except for mass unit, for which gram was used instead of kilogram). In the simulations, the model was operated with an hourly time

step for outdoor emissions and a daily time step for indoor emissions with corresponding units being converted.

#### 2.1.1 Mass balance of nitrogen components

The AMCLIM-Poultry model simulates masses for N-containing components (UA, TAN) within the chicken farming system (chicken houses; backyard chickens; and chicken manure spreading), and flows between these pools (Fig. 1). The mass per unit area of excretion ($M_{excretion}$, g m$^{-2}$; all model variables are described, with units, in the Appendix) is calculated following

Eq. (1):

$$M_{excretion}(t + \Delta t) = M_{excretion}(t) + \frac{F_e}{f_N} \Delta t, \tag{1}$$

where $F_e$ (all nitrogen flows have units of g N m$^{-2}$ s$^{-1}$) is total nitrogen excretion rate from chicken and $f_N$ (g N g excretion$^{-1}$) is the nitrogen content of excretion. The evolution of UA mass ($M_{UA}$; all nitrogen pool masses have units of g N m$^{-2}$) over time-step $\Delta t$, is calculated following Eq. (2):

$$M_{UA}(t + \Delta t) = M_{UA}(t) + (F_e f_{UA} - F_{TAN})\Delta t, \tag{2}$$



where $f_{UA}$ is the UA fraction in the excretion, and $F_{TAN}$ is the conversion rate of UA to TAN.

Similarly, the mass of TAN ($M_{TAN}$) is calculated following Eq. (3):

$$M_{TAN}(t + \Delta t) = M_{TAN}(t) + (F_{TAN} - F_{NH_3})\Delta t, \tag{3}$$

where $F_{NH_3}$ is the net rate of conversion of TAN to gaseous $NH_3$ that is emitted to the atmosphere.

**2.1.2 Process-based simulation of nitrogen pathways**

For each emission context (i.e., animal housing, backyard birds, manure spreading), the AMCLIM-Poultry model includes three key steps: conversion of UA to TAN, equilibrium between aqueous phase TAN and gaseous $NH_3$ in the litter, and volatilization of $NH_3$ from the litter surface to the atmosphere (Fig. 2). Uric acid is converted to TAN by hydrolysis, which is strongly affected by temperature, the pH of the substrate, and the relative humidity (RH) of the chicken house atmosphere

(Elliott and Collins, 1982; Elzing and Monteny, 1997; Koerkamp, 1994). The production rate of TAN is determined from the UA mass and the conversion rate ($K$), which is a function of these three factors:

$$F_{TAN} = M_{UA}K_{(T,pH,RH)} \tag{4}$$

Gas phase $NH_3$, held within the litter pore spaces, is in equilibrium with TAN that depends upon the litter pH and temperature response of combined Henry and disassociation equilibria (Eq.(5)) (Nemitz et al., 2000). The gas phase concentration of $NH_3$

in air ($\chi$) at the surface is proportional to the aqueous phase ratio $\Gamma = [NH_4^+]/[H^+]$ of the chicken litter, which is calculated from Eq. (5) and Eq. (6):

$$\chi = \frac{161500}{T} \exp\left(\frac{-10378}{T}\right) \Gamma, \tag{5}$$

$$[NH_4^+] = \frac{M_{TAN}}{V_{H_2O}}, \tag{6}$$

where $[NH_4^+]$ is in units of g N ml$^{-1}$ and $V_{H_2O}$ (ml m$^{-2}$) is the volume of water in the litter. Ammonia volatilises to the

atmosphere from the surface at a rate ($F_{NH_3}$) that can be determined by assuming a resistance type model: using gas concentrations at two vertical levels constrained by a set of resistances (Sutton et al., 2013), which is calculated from Eq. (7):

$$F_{NH_3} = \frac{[\chi(z_{o'}) - \chi(z)]}{[R_a(z) + R_b]}, \tag{7}$$

where $\chi(z_{o'})$ represents the concentration at the surface, and $\chi(z)$ represents the concentration at a reference height. Equation (7) is the general formula. For in-house application of the model, $\chi(z)$ is taken as representative of well mixed indoor

concentration of $NH_3$ in chicken house. For outdoor application of the model, the reference height is taken 10 m above ground. $R_a$ and $R_b$ are the aerodynamic and boundary layer resistances, respectively. This broad resistance approach is applicable for manure spread in the field and is also applied for backyard birds. For emissions from housed chicken a modified approach is needed as described in Sect. 2.2.2.



## 2.2 Simulations for chicken housing

Figure 2 illustrates the process pathways through which $NH_3$ volatilises from the N-rich chicken excretion to the exterior atmosphere. We assumed 60 % of excreted nitrogen is in the form of UA ($f_{UA}$ = 0.6), which accounts for approximately 3-8 % of the chicken excretion (Nahm, 2003). The remaining 40 % of excreted nitrogen is in all other forms that are not liable to significant $NH_3$ emissions. Uric acid accumulates in the litter of the chicken house until it converts to TAN by bacterial ammonification, with TAN concentrations in equilibrium with the litter pore space concentration of gaseous $NH_3$. Ammonia is then emitted from the surface, which builds up the indoor $NH_3$ levels within the house through mixing. Meanwhile, $NH_3$ is removed continuously through ventilation because the indoor $NH_3$ concentration must be controlled below a certain level.

We used the monitored data from Animal Feeding Operations (AFOs, 2012) to simulate site-specific $NH_3$ emissions from chicken houses. The data were gathered by the US Environmental Protection Agency (EPA) as a study of emissions from different types of livestock from 2007-2010 (Cortus et al., 2010; Jin-Qin Ni et al., 2010; Wang et al., 2010). As shown in Table S1, two broiler houses and four layer houses from three US farms at different sites were selected for this study. We used daily mean animal data, environmental data, and indoor $NH_3$ concentrations (measured at 2 - 2.5 m above the ground, representative of well mixed air in the chicken house) from these sites. Animal data included bird numbers, body weight, and biomaterial data for each house, and environmental data included temperature, relative humidity for natural (outdoor) and indoor conditions, and the interior ventilation given as an airflow rate in $m^3$ $s^{-1}$. We filled up missing environmental data by using a linear interpolation method when measurements were unavailable to keep simulations continuous. Excreted nitrogen was determined from the animal data and was used as an input to the model, together with the indoor environmental data. The excretion water content ($M_{H_2O}(e)$, g $m^{-2}$) that determines the TAN concentration of litter is dependent on the equilibrium moisture content ($m_E$, %) of the litter, which is calculated from Eq. (8):

$$M_{H_2O}(e) = \frac{m_E}{100} \cdot M_{excretion}, \tag{8}$$

where $m_E$ is calculated following the Eq. (9):

$$m_E = \left[ \frac{-\ln\left(1 - \frac{RH}{100}\right)}{0.0000534 \times T} \right]^{\frac{1}{1.41}}, \tag{9}$$

where $RH$ (%) is the relative humidity, and $T$ (K) is the temperature (Elliott and Collins, 1982)

### 2.2.1 Parametrization of UA hydrolysis rate for chicken housing

The hydrolysis of UA to TAN plays a crucial role in affecting $NH_3$ emissions. The rate of conversion of UA to TAN is often the rate-limiting process that determines the overall rate of conversion of nitrogen excreted by chicken into $NH_3$ emissions. The parametrization of UA to TAN conversion is therefore very important for the overall model performance.

In the study of Elliott and Collins (1982), a chicken litter model was used to investigate the UA hydrolysis rate. They set the base level conversion rate to 20 % over a 24-hour period under optimal conditions (pH = 9, T ≥ 35 °C, RH ≥ 80 %), then produced empirical functions to account for the influence of these three factors. In order to evaluate the validity of these empirical functions, specifically temperature and RH effects, we analysed the AFO measurements for two layer houses from the US EPA dataset (Table 1), starting from the date that litter was cleaned out from the houses. We assumed an equilibrium





state between the production of TAN and NH₃ emission, and a constant litter pH of 8.5. It should be noted that the equilibrium state does not always apply, but it is a useful assumption for parameterization, and the introduced uncertainty is discussed in Sect. 4.1.2. The temperature dependence was derived from measurements when RH was over 80 %, and the RH dependence was derived from measurements that were normalised by the temperature dependence. We used these data to update the

empirical functions of Elliott and Collins (1982) that parameterize the UA hydrolysis rate (see Sect. 3.1.2).

**2.2.2 Inversion of resistance within chicken houses to develop R\* parametrization of chicken houses**

The NH₃ flux from an unvegetated surface to the atmosphere is mainly constrained by two terms: aerodynamic resistance ($R_a$) and boundary layer resistance ($R_b$) (Wesely, 1989). Outdoors, both these resistances are related to meteorological conditions and can be calculated. However, values of $R_a$ and $R_b$ within chicken houses remain unknown due to the lack knowledge of

turbulence for indoor conditions. We estimated the overall indoor resistance, termed $R^*$, by inversion of the measured AFO data. As shown by steps 4, 5 and 6 in Fig. 2, the interior NH₃ level within a chicken house is determined by the source flux from the litter surface and the removal flux through ventilation. Mathematically, the total flux of NH₃ ($F_{surface}$, g N s⁻¹) from the surface is expressed as Eq. (10):

$$F_{surface} = \left(\frac{\chi_{surface} - \chi_{in}}{R^*}\right) \cdot S, \tag{10}$$

where $\chi_{surface}$ (g m⁻³) is the in-house value of $\chi(z_{o'})$, i.e, the gaseous NH₃ concentration at the litter surface and $\chi_{in}$ (g m⁻³) is the indoor NH₃ concentration of the house assuming a complete mixing of air inside the chicken house. $R^*$ (s m⁻¹) is the indoor resistance, and $S$ (m²) is the surface area of the house. The NH₃ removal ($F_{removal}$, g N s⁻¹) through ventilation is expressed as Eq. (11):

$$F_{removal} = Q\left(\chi_{in} - \chi_{out}\right), \tag{11}$$

where $\chi_{out}$ (g m⁻³) is the free-atmosphere NH₃ concentration. $\chi_{out}$ is set to be 0.3 μg m⁻³, which is normally much lower than the indoor concentration. $Q$ (m³ s⁻¹) represents the ventilation rate. Therefore, by mass conservation, we can relate indoor NH₃ concentrations and the interior air volume $V$ (m³), to surface emissions and losses through ventilation:

$$V\frac{d\chi_{in}}{dt} = F_{surface} - F_{removal}$$

$$= \left(\frac{\chi_{surface} - \chi_{in}}{R^*}\right) \cdot S - Q\left(\chi_{in} - \chi_{out}\right) \tag{12}$$

For inversion of $R^*$, we used the data for two layer houses at NC2B, which had clearly reported house emptying dates and had fewer missing measurement data. The simulation period started from the day when litter was cleaned out, and each nitrogen pools was re-initialised. For the inversion, we assumed the house reached steady-state (hence the LHS of eq. (12) is zero) after a period of simulation for three days. Subsequently, the resistance can be calculated from Eq. (13):

$$R^* = \frac{(\chi_{surface} - \chi_{in}) \cdot S}{Q\chi_{in}} \tag{13}$$

To develop this parametrization, the gas phase NH₃ concentration at the surface ($\chi_{surface}$) was simulated by the AMCLIM-Poultry model and the NH₃ concentration within the house and ventilation were taken from the AFOs monitored data.



### 2.3 Simulations of NH$_3$ emission from chicken manure spreading

Contrary to the housing, the simulation of NH$_3$ emissions from the spreading of chicken manure to fields is different due to the following points. First, the amount of water in the system ($M_{H_2O}$, g m$^{-2}$) is related to the outdoor environment (i.e. precipitation, evaporation and runoff):

$$M_{H_2O}(t + \Delta t) = M_{H_2O}(t) + M_{H_2O}(e) - M_{available\,water} + (F_{H_2O}(rain) - F_{H_2O}(evap))\Delta t, \tag{14}$$

where $F_{H_2O}(rain)$ (g m$^{-2}$ s$^{-1}$) and $F_{H_2O}(evap)$ (g m$^{-2}$ s$^{-1}$) are the precipitation and evaporation, respectively, and $M_{available\,water}$ (g m$^{-2}$ s$^{-1}$) is the water available for run-off. It should be noted that the amount of water of the system should not be less than the excretion water content. The maximum amount of water that can be absorbed by the manure, which was assumed to be a factor of 2 × of the mass of excretion (Riddick et al., 2017). The water left in the system is the amount of water available for runoff ($M_{available\,water}$, g m$^{-2}$):

$$M_{available\,water} = F_{H_2O}(rain)\Delta t - 2 \times M_{excretion} \tag{15}$$

Second, runoff takes place under natural conditions especially during rain events and is a major loss of nitrogen. In the model, the immediate runoff ($M_{N\text{-}runoff}$, g m$^{-2}$) is derived from the runoff coefficient multiplied by the nitrogen pools:

$$M_{N-runoff} = R_{runoff} \cdot M_N, \tag{16}$$

where the $M_N$ (g m$^{-2}$) is the amount of each N-containing components, and $R_{runoff}$ is the runoff coefficient that is a function of the amount of water within the nitrogen pools available for runoff ($Q_{available\,water}$, mm):

$$R_{runoff} = Q_{available\,water} \cdot r_N, \tag{17}$$

where $r_N$ (mm$^{-1}$) represents the wash off factor, and constant values was used of 1 and 0.5 % mm$^{-1}$ for nitrogen and manure, respectively (Riddick et al., 2017).

Third, the resistances including aerodynamic ($R_a$) and boundary layer resistance ($R_b$) were directly calculated from meteorological variables instead of being parameterized (Nemitz et al., 2001; Seinfeld and Pandis, 2016; Riddick et al., 2017).

### 2.4 Global applications

### 2.4.1 Model input

In order to quantify the NH$_3$ emission from global chicken farming, we applied the AMCLIM-Poultry model at the global scale. The model used the FAO (Food and Agricultural Organization of United Nations) global chicken density data and chicken excretion nitrogen data as input and was driven by the ECWMF ERA5 hourly meteorological data (ERA5, 2018). The model was run under a resolution of 0.5° × 0.5°, with the global chicken density data and nitrogen data being regridded to fit the 0.5° resolution.





The global population of chickens was based on FAOSTAT data for 2010 (FAOSTAT). The geographic distribution was based on the Gridded Livestock of the World (GLW) model, which produced density maps for the main livestock species based on observed densities and explanatory variables such as climatic data, land cover and demographic parameters (Robinson et al., 2014). The chicken data were categorised into three production systems: broilers, layers and backyard chicken. Broilers and

layers are major chicken types that are reared intensively in buildings and managed by farmers or livestock companies. The environment for rearing backyard chicken is varied and the density is lower compared with broilers or layers. The distinction in the global distribution of backyard and intensive systems was based on Gilbert et al. (2015). Birds in the intensive systems were further subdivided into broilers and layers using the procedure developed for the Global Livestock Environmental Assessment Model (GLEAM FAO, 2018). The GLEAM approach was also used to produce the nitrogen excretion maps,

which were calculated as the difference between nitrogen intake and retention. The total nitrogen intake depends on feed intake and nitrogen content of the feed, while the retention is the amount of nitrogen that is retained in birds' tissues, either as live weight gain or production of eggs (FAO, 2018).

### 2.4.2 Global upscaling for chicken housing

In chicken farms, the inside conditions can be distinct from the natural environment. The 'lower critical temperature' for

chicken (i.e., the minimum managed temperature for optimum chicken performance) is approximately 16-20 °C (Gyldenkærne et al., 2005) which is much higher than of other livestock, such as cattle and sheep. Intensively managed chicken are typically kept in insulated buildings with forced ventilation and heating systems to help maintain fixed temperature throughout the year as far as feasible (Seedorf et al., 1998). To keep the ambient temperature within a recommended range, the house may be heated or ventilated in relation to outdoor temperatures. Heating occurs on cold days when temperature is low but not in other

periods. Ventilation is to maintain a healthy condition for chicken's growth, and a minimum level is required, but also the ventilation should be below a certain rate to avoid induced draft in the house (Gyldenkærne et al., 2005).

For the modelling, the broilers and layers were assumed to be kept in buildings with adequate heating and ventilation systems. The density for broilers and layers was assumed to be 15 birds/m$^2$ and 30 birds/m$^2$, respectively (Cortus et al., 2010; Jin-Qin Ni et al., 2010; Krause and Schrader, 2019; Wang et al., 2010). In the AMCLIM-Poultry model, the environmental parameters

incorporated in the model are empirically derived from the indoor environment of chicken farms reported in the EPA dataset. It is assumed that the temperature and ventilation rates of houses are maintained as close as possible to a stable level throughout the day and are driven by the natural climatic conditions under local practice. There is no precipitable water in the house, so the water budget excludes precipitation and is determined by excretion moisture. The litter in chicken houses was assumed to be removed once a year. The housing part of the AMCLIM-Poultry model was operated at a daily time-step based at 2010. 12

simulations were run by assuming that chicken houses were emptied in different months for each simulation, i.e. from January to December, and the simulations started in corresponding month. The results were averaged and reported in this study.

### 2.4.3 Global upscaling for chicken manure spreading

As shown in Fig. 1, the manure from chicken farms are collected for applications to fields, leading to NH$_3$ emissions. Typically, fertilising crops use manure from local farms. Therefore, we assumed the amount of nitrogen from chicken manure is only

spread locally, and the simulations for each grid-cell are independent to the adjacent ones in terms of model input. This assumption is considered to be valid at 0.5° × 0.5° resolution of the global model application (equivalent to 39 km × 55 km at 45° latitude), though cannot be automatically assumed when modelling at finer scales. The available nitrogen budgets were determined from the amount of nitrogen left, ensuring mass-consistency to account for NH$_3$ emitted in the housing simulations.



It should be emphasized that the global distribution of available nitrogen for land spreading of chicken manure may not completely coincide with global distribution of croplands or the global usage of inorganic nitrogen fertilisers. It is assumed in the model application here that chicken manure is only used on arable lands, so there should not be any manure applications from intensively managed housed chicken regions with no farming practice. Meanwhile, there are thresholds for nitrogen

applications for crops. If nitrogen application rates required to use the chicken manure on agricultural land exceed the maximum guided amount, it will have harmful or lethal effects on crops. Therefore, simply using the total available nitrogen from livestock manure as inputs to the land spreading part of the AMCLIM-Poultry model could cause error and not reflect reality.

To address these considerations, we defined the amount of nitrogen applied to crops as contributed nitrogen input. To estimate

the contributed nitrogen input from chicken manure, we compared the available amount of chicken manure-N (nitrogen left in manure after being lost as $NH_3$ at housing period) to the total amount of manure-N for crops to identify places that would use chicken manure as fertiliser. Data of the total amount of manure-N used for crops and fertilising areas were used from West et al (2014). We chose six major crops for which chicken manure is ideal fertiliser, including barley, maize, potato, rice, sugar beet and wheat. We assume the chicken manure is primarily applied to these six crops. For areas where available chicken

manure-N does not exceed the total manure-N application, we calculate the contributed nitrogen input for individual crops by Eq. (18):

$$N_{Crop\_Poultry} = N_{Soil\_Poultry} \cdot \frac{N_{Crop}}{N_{Total\_Manure}}. \qquad (18)$$

Conversely, for areas where available nitrogen input from chicken exceeds the total manure-N application, the contributed nitrogen input is calculated from Eq. (19):

$$N_{Crop\_Poultry} = N_{Crop}, \qquad (19)$$

where $N_{Crop\_Poultry}$ (g N m$^{-2}$) is the amount of chicken manure-N application for individual crops, $N_{Soil\_Poultry}$ (g N m$^{-2}$) is the amount of available chicken manure-N, $N_{Crop}$ (g N m$^{-2}$) is the amount of total nitrogen application for individual crops, $N_{Total\_Manure}$ (g N m$^{-2}$) is the amount of total nitrogen application for all crops. The excess nitrogen in these areas was considered to be applied to other crops. In regions where annual nitrogen applications are zero, we assumed the available chicken manure-

N are untreated and left on land.

Planting and harvesting dates for crops are important parameters in the model because they determine the meteorological conditions of the crop growing period, which affects the temporal variations of $NH_3$ emission from land spreading. Fertiliser applied to land or crops is dependent on the timing of agricultural activities rather than being spread frequently. As a result, the $NH_3$ emission from fertiliser spreading usually shows strong seasonal variations due to the local farming practice. In this

study, the model incorporates the planting and harvest dates from the Crop Calendar Dataset for the six major crops to make estimates (Sacks et al., 2010). We developed a relatively simple scenario for fertiliser applications that the chicken manure fertiliser was applied at the start of planting period. Timing of agricultural practices in the southern hemisphere is different from the northern hemisphere. The planting activities usually start in November or December, which causes that partial $NH_3$ emissions in these regions would occur in the next year. Similarly, manure spreading that took place in the last year can also

result in emissions in the current year. Therefore, we ran the model for more than one year to keep an annual cycle of simulation period for each grid. It should be emphasized that our model scenario assumes a standard reference that all chicken manure is



broadcast on the surface of bare agricultural fields, at the start of the cropping cycle. Other future scenarios could consider the effectiveness of management practices to mitigate NH₃ emission from the spreading of chicken manure (see Sect. 4.4).

As introduced in Sect. 2.4.1, backyard chicken is one of the major production systems included in the FAO chicken density dataset. In comparison with broilers and layers, backyard chicken is reared in residential lots rather than in insulated houses.

According to the FAO statistics, there are two general ways of dealing with excretion from backyard chicken: daily spreading and leaving it on pastures. Consequently, the simulations for NH₃ emissions from backyard chicken were set to be under natural environments. Data for excreted nitrogen from backyard chicken from the FAO dataset were used as the nitrogen input to the model. The density was assumed to be 4 birds/m². The meteorological inputs were the same as used in the simulations for chicken manure spreading for crops. The model was operated at an hourly time-step for a period of one year as an

initialisation. The second-year simulation was for the study period of 2010.

## 3 Results

### 3.1 Site simulations for chicken housing

#### 3.1.1 Temperature of chicken houses

A generalised representation of indoor temperatures of chicken housing was empirically derived from the AFOs measurements

from the three farms. The relationships between indoor temperature and outdoor temperature of broiler houses and layer houses are different (Fig. S1). In layer houses, temperature is considered to be primarily dependent to the outdoor temperature, while broiler houses' temperature is also related to broilers' body weights, as these range from chicks to harvested adults and as special conditions are typically applied for chicks. Chicks are typically reared under relatively warm conditions, with the temperature around 32-35°C. However, NH₃ emission at this stage is tiny because the nitrogen excretion rate of chicks is low,

and litter is typically fresh. For broilers, NH₃ emission mostly takes place from the later growing period once excretion rates are larger and litter has built up. Based on the measurements from animal house CA1B (Table S1), the indoor temperature of broiler housing was taken into account (as shown in Fig. S1) for the period in which the body weights exceed a threshold of 0.5 kg.

#### 3.1.2 Factors affecting UA hydrolysis rate

Decomposition of UA from chicken excretions to TAN is dependent on the temperature, moisture, and pH of the substrate. The maximum estimated breakdown rate is 20 % per day at 35 °C, pH 9.0, and RH 80 % (Elliot and Collins, 1982). The combined influence of three factors is the product of a series of conversion rate functions as expressed by the Eq. (20).

$$K_{(T,pH,RH)} = 0.2\, k_{pH} k_T k_{RH} \tag{20}$$

We used the pH dependence for the range of 5.5~9.0 from the Elliott and Collins (1982) study:

$$k_{pH} = \frac{1.34(pH) - 7.2}{1.34\,(9) - 7.2} \tag{21}$$



The temperature and RH dependence of UA hydrolysis rate derived from using the AFO monitored data are shown in Fig. 3, where they are compared to functions from Elliott and Collins (1982). The new temperature dependence follows an exponential relationship, and is normalised to the maximum rate at 35 °C:

$$k_T = \frac{exp^{(0.149(T-273.15)+0.49)}}{exp^{(0.149(35)+0.49)}} \tag{22}$$

The new RH dependence increases linearly as RH increases, reaching the maximum rate of 1 at RH 80 %:

$$k_{RH} = 0.0124\, RH - 0.0014 \tag{23}$$

It should be noted that the RH dependence within the range of RH 0~40 % is extrapolated because there were limited data at these conditions (Fig. 3b).

### 3.1.3 Resistance within chicken houses and site simulations

The inversion derived resistance within chicken houses, R*, is presented in Figures S2 to S5; strong daily variations can be seen. The possible relationships of calculated R* values to temperature and ventilation rate were investigated. This showed no strong correlation with these indoor environmental variables (See Fig. S6 and Fig. S7). We simulated the total $NH_3$ emissions with various constant R* values throughout the year and compare the results to the measurements (Fig. S8). A fixed R* value of ~ 16700 s m$^{-1}$ was found to provide the best result of 1:1 for House A, and ~ 14369 s m$^{-1}$ for House B at NC2B.

Figure 4 and 5 show the simulated indoor $NH_3$ concentrations and emissions comparing to the measurements by assuming the fixed R* value of 16700 and 14369 s m$^{-1}$, respectively. Gaps occurred in measured $NH_3$ concentration and emissions were due to unavailable measurements, while the model was kept running. The model was able to capture the major changes throughout the simulation period. During hot periods of the year, the temperature inside the house was generally higher than cold months, and ventilations rates reached the maximum. High temperature led to large UA hydrolysis to increases the TAN pool, which 20 allows more $NH_3$ emissions. High ventilation rates accelerated the $NH_3$ removal from the house, and the indoor concentration of $NH_3$ decreased. The TAN pool of both houses accumulated and reached approximately 5 kg m$^{-2}$, while the UA pools were relatively low due to the continuous conversion to TAN. Sharp declines of the UA pools were seen (dates April/09/2008 in House A, June/03/2008 in House B), linked to the chicken houses being empty at these times (as shown by black dash lines) for approximately three weeks. As a result, with sufficient TAN and large difference between surface and air $NH_3$ 25 concentration, $NH_3$ emissions in hot months were high.

### 3.1.4 Model sensitivity to temperature and relative humidity

To understand the effects of temperature and relative humidity on the $NH_3$ volatilization in chicken houses, we ran simulations under idealised conditions. We used a configuration (i.e. animal number, house size) the same as the NC2B House A, but set the temperature and relative humidity to constant values throughout the whole year. A spin-up year run was prior to the 30 experimental simulations.

We tested the $NH_3$ volatilization rate ($P_V$) under a domain with temperature range of 15-35 °C and RH range of 20-100 %. Figure 6 shows an overall increasing of $P_V$ from low temperature and RH to high temperature and RH regime. The highest $P_V$ values reaching approximately 56 % were from high temperature and RH simulations. Figure 7a shows that the $P_V$ rates



increase as temperature increases, and Fig. 7b also shows that the $P_V$ rates increase as RH increases, but drop after RH exceeds 90 %.

### 3.2 NH₃ emission from global chicken housing

We used the polynomial fits shown in Fig. S1 and the constant R* values of 16700 s m$^{-1}$ as representative of all chicken houses

for the simulation of global emissions. The estimate of NH₃ emission from global chicken housing in 2010 was 2185.5 Gg N. This includes 1374.7 Gg N emissions from broilers and 810.8 Gg N from layers. Figure 8 shows high emissions in Europe, India, China and Southeast Asia, with emission hotspots in eastern US, and the eastern part of South America. The total amount of nitrogen from chicken excretion was 9017.1 Gg in 2010. The percentage of nitrogen excreted that is volatilized as NH₃ ($P_V$, %) was estimated at 24.2 % overall for all NH3 emissions from chicken housing globally. The value $P_V$ for chicken housing

was high across the tropics, reaching approximately 35 % (Fig. 8b). Regions with high NH₃ emission mostly show high NH₃ volatilization rates, especially in regions such as east China, Southeast Asia, and east US. As the $P_V$ value normalizes for chicken numbers, it more clearly shows the influence of climate than total NH₃ emissions. Figure 8b shows very small $P_V$ values in dry areas (Sahara, Australia, Arabian peninsula, Patagonia, Central Asia, western North America, illustrating low humidity in these areas is estimated to limit UA hydrolysis, with the converse in humid areas (Amazonia, central Africa, south

east Asia, etc).

### 3.3 NH₃ emission from global chicken manure spreading

### 3.3.1 NH₃ emission from chicken manure application for crops

For the year 2010, the NH₃ emission from chicken manure application for crops was 2582.3 Gg N, with the $P_V$ value representing 37.8 % of the total nitrogen application to land of 6827.0 Gg N. The nitrogen considered to be left untreated

according to Sect. 2.4.3 was 4.6 Gg, which is only a small fraction compare to the amount of nitrogen applied to land. From simulations in this study, over 75 % of the NH₃ emissions were from applications for the major 6 crops specified in Sect. 2.4.3, while the rest were from applications for other crops (Table S2). Among the 6 crops, maize fertilising contributed to the highest emission of 643.4 Gg N, which is more than 1/3 of the total amount. Fertilising rice and wheat also led to 601.4 and 520.3 Gg N of emissions, respectively. Compared with maize, rice and wheat, crops of barley, potato and sugar beet had much smaller

emissions due to lower estimated total application of chicken manure to these crops (reflecting their smaller cropping areas and the chicken distribution). The NH₃ volatilization of crops all six crop types exceeded 34 % (Table S2). The application for rice resulted in the highest $P_V$ of 42.0 %, (reflecting the warm and moist climate of rice cropping), while the application for barley had the lowest $P_V$ values of 34.5 % (reflecting its distribution in cooler temperate climates).

The geographical distribution of NH₃ emissions from chicken manure application is presented in Fig. 9a. Similar to the chicken

housing, high emission can be seen in Europe, eastern Middle East and south India, while extremely large NH₃ emission exceeded 10 Gg N yr$^{-1}$ over eastern and central part of China and south east Asia, with hotspots in south eastern US, Mexico and eastern South America. These hotspots reflect a combination of high chicken populations and high $P_V$ values. Areas of the lowest $P_V$ are associated with cropping areas having the lowest rainfall, including west central North America, southern Africa and central Asia. Areas estimated to have no significant arable cropping (i.e., desert, boreal and tundra) are shown white

in Fig. 9.





### 3.3.2 NH₃ emission from backyard chicken

The global NH₃ emission from backyard chicken in 2010 was estimated at 714.5 Gg N from a total excreted nitrogen of 2178.3 Gg. Backyard chicken density showed a different distribution compared with broilers and layers (Fig. S10). This reflects the assessment in the FAO database that backyard chickens are not kept in developed countries including Canada, United States

of America, west Europe, Australia and New Zealand, where all chicken are allocated to housed systems. The FAO database estimates that most backyard chicken occur in developing regions, such as the northern India and Africa. Geographically, the highest emission from backyard chicken are here estimated to occur in Ukraine, south and south-east Asia, with high emissions in east coastal regions of South America and the southern part of West Africa. Figure 10b illustrates the geographic distribution of the percentage nitrogen volatilized ($P_V$). The volatilization rates of vast majority of Asia were less than 24 %, while the

tropics including South Asia had higher $P_V$ rates that reach 36 %. Possible reasons for the different distribution of $P_V$ for backyard birds as compared with manure application to crops are discussed in Sect. 4.2.

### 3.4 Annual NH₃ emission inventory for global chicken farming

The estimated NH₃ emissions based on 2010 are summarised in Table 1, and the geographic distribution is presented in Fig. 11. Overall, the total emission from global chicken farming was 5482.3 Gg N yr⁻¹. Practice related to broilers and layers

including housing and manure application to crops contributed 1179.6 and 3372.9 Gg N NH₃ emissions, respectively, and backyard chicken manure caused 714.5 Gg N emissions. Regions with high NH₃ emissions were across Europe, India, and part of China, with hot spots occurred in East US and Eastern South America. The distribution of $P_V$ values reflects the combined effect of how environmental differences lead to variations in emissions from chicken housing, manure spreading to arable land and from backyard birds.

Figure 12 shows the NH₃ emissions from the three main components for chicken (housing, crops, backyard) and the corresponding volatilization for 5 latitudinal bands. The highest emission was between 20 ~ 40 °N, reaching a total NH₃ emission of 2540.8 Gg N. The lowest emission was 317.2 Gg N between 20 ~ 40 °S. Manure application to crops was the largest fraction of NH₃ emissions in the northern hemisphere, and its volatilization to NH₃ was the highest among the three categories across the globe, exceeding 35 %. The NH₃ volatilizations of housing and backyard chicken were comparable,

ranging between 20 % to 30 % of the total emission. Figure 12 summarizes the latitudinal difference in percentage volatilized. The smaller degree of variation reflects the complex way in which water availability, humidity and temperature interaction to affect the overall percentage of nitrogen volatilized, as illustrated by the maps.

Figure 13a shows the monthly NH₃ emissions from each sector. Highest emissions of over 600 Gg N were estimated for April and August, while lowest estimated emissions were in November, December and January. This shows how the seasonal

differences are larger for NH₃ emissions from manure application to crops than from animal houses, which is a result of both the climatic effects, and the temporal distribution of manure application according to the start of the main cropping seasons. From Fig. 13b, the NH₃ volatilization from backyard chicken excretion varied more throughout the year than for housing (linked to larger variations in temperature and water availability). Emissions from backyard birds were higher than housing from April to August, with the largest difference in July, and were lower than housing from September to March. The highest

estimated rate was 65.4 % in July and lowest rate was 12.2 % in January. The volatilization rates of housing showed smaller variations, with $P_V$ values mostly over 20 %, with the highest rate of 30.9 % occurring in August. It is worth noting that volatilization rates of manure land spreading are not presented in the figure because simple monthly values do not reflect the





true volatilization rate. Nitrogen being applied in the agricultural month will cause $NH_3$ emission in the following months when no application practices take place.

## 4 Discussion

### 4.1 Parametrisations for chicken housing

**4.1.1 Indoor environmental conditions of chicken houses**

Meteorological conditions affect the $NH_3$ emissions from chicken housing indirectly by influencing the indoor environmental conditions, which is crucial in affecting $NH_3$ volatilization. At high temperatures, the ventilation rate is increased to cool down the house, keeping the inside temperature close to the reference value. When ventilation systems reach their maximum flows, indoor temperature would continue to rise above the reference. Increasing ventilation rates help minimize temperature

increases, increase water evaporation of the house, and reduce the moisture associated with chicken excretion. Theoretically in warm dry conditions, net $NH_3$ emission tends to decline because of the less efficient UA hydrolysis. By contrast, in humid conditions, increased ventilation of chicken houses under warmer conditions is estimated to increase $NH_3$ emissions in the model, as UA hydrolysis is favoured and $NH_3$ are quickly removed from the houses to the atmosphere.

It is worth noting that management for broiler rearing is different from layers. The growth period of which broilers from chicks

to adults is approximately 6-8 weeks. At initial stage, the house is heated to keep the inside temperature up to 32-35 °C, allowing the chicks to grow under a warm and comfortable condition. As the birds are growing stronger and gaining weight, the indoor temperature is decreased. Once the adult birds gain enough weight, they are removed from the house. The house is then empty for the next 3-7 days until another flock is settled in, and the heating system is turned off. In comparison, egg layers are kept longer in houses, which normally lasts for over 2 years. The indoor temperature of a layer house is controlled, as far

as possible, within a referenced range throughout the year. The manure management also varies. According to the AFO's dataset, broiler houses in the US are cleaned after every 3-4 flocks, and the excretion with litter or bedding materials removed, while layer houses are usually designed to have multiple floors, allowing the litter to be collected and removed at the lower floor by conveyor belts. These differences have implications for $NH_3$ emissions between broiler and layer systems, the most important one being the need to recognize the cycles of temperature and humidity as these affect $NH_3$ emissions from broilers.

Even if litter is not cleared out after removing grown broilers, it is anticipated that new bedding material will be added, therefore covering the old litter, so that emissions are mainly related to the excretion of each flock. Since most emissions are associated with older broilers, this has allowed the simplification (Sect. 3.1.1), that the relationship between indoor and outdoor temperatures is based on periods where birds are >0.5 kg. While the relationship between indoor and outdoor temperatures applied here is based on the US EPA experimental farms, access to such datasets for other climates would be useful to extend

and improve the parametrization.

### 4.1.2 Comparison between the empirical equations for UA hydrolysis

Figure 3 shows the parameterizations for UA hydrolysis in chicken houses from this study and the Elliott and Collins (1982) study. The temperature dependences are comparable in that both studies suggest an exponential correlation between the Factor T and indoor temperature. Overall, the Factor T derived from using the AFOs monitored data in this study was slightly larger

than that from Elliott and Collins (1982). Within the temperature range of 18 to 28 °C, the UA hydrolysis rate approximately doubled every 5 °C, and an increasing 10 °C led to more rapid hydrolysis rate by a factor of 4.4 and 5.2 based on the two





studies, respectively. In contrast, the RH dependences were more different between the two studies. The new parameterization suggests a linearly decline of Factor RH as RH decreases below 80 %, so that the magnitudes of Factor RH are much larger compared with Elliot and Collins (1982). When RH is below 40 %, the Factor RH for the present study was obtained from extrapolation due to the lack of measurement from the AFOs dataset.

The results of global simulations by using two parameterizations are presented in Fig. 8 (using RH parametrization from Elliot and Collins, 1982) and Fig. S9 (using the RH parametrization based on Fig. 3 from the monitored AFOs). The annual $NH_3$ emissions from housing in 2010 were estimated at 3312.4 Gg N based on the new parameterization (from the monitored AFOs), giving 51.6 % higher emissions than the estimates of 2185.5 Gg N using the equations of Elliott and Collins (1982). In principle, warmer and wetter conditions lead to an increase in $P_V$. Increasing temperature accelerates the formation of TAN

and increases the surface concentration of $NH_3$, and the hydrolysis of UA is enhanced under high moisture environments. The temperature inside chicken houses in the AMCLIM-Poultry model is assumed to be controlled, especially the houses in cold climate regions, where sufficient heating is assumed to be used to maintain healthy environments. Therefore, the variations of housing temperature were not as significant as the outdoor temperatures. On the other hand, the houses prevent the rain getting in, so the hydrolysis of UA and aqueous $NH_3$ concentration are solely restricted by the water content of the excretion, which

is a function of RH. As a result, RH becomes the foremost factor that determined the $NH_3$ emissions by affecting the water availability of the system. It is notable that large differences between the two sets of global simulations (as shown in Fig. 8 and Fig. S9) occurred in dry regions, such as Northern Africa, the Middle East, and Western Australia. Compared with the results of using the Elliott and Collins equations, the new parameterization suggests much higher $NH_3$ volatilization in dry places. The substantial difference between the model simulations using the two RH parametrizations indicate the need for

further data on this relationship. Additional measurement datasets including both temperature and RH measurements, and representing a wider range of environmental conditions, would help to strengthen and extend the relationships observed.

It must also be recognized that both the RH parametrizations shown in Fig. 3b have limitations. A more accurate parameterization of RH dependence might fall in the area between two curves in Fig. 3b. It can be seen from Fig. 4c and Fig. 5c that the TAN pool of each chicken house increased continuously throughout the simulation period rather than remaining

approximately constant at some points. This indicates that the TAN produced exceeded the loss through $NH_3$ emission, which is against the assumption that the production of TAN is equivalent to the $NH_3$ emission. It is possible that this overestimated the rate of UA hydrolysis. Meanwhile, from the Fig. S4 and Fig. S5, by using Elliott and Collins's parameterization for RH dependence of UA hydrolysis, the modelled indoor concentration of $NH_3$ was much lower than the measurements during the starting period of simulations. This was caused by the insufficient TAN pool that limited the emissions. Therefore, Elliott and

Collin's parameterization probably underestimated the TAN production from UA hydrolysis, especially when each nitrogen pool was limited. In addition to the need for further datasets that relate $NH_3$ emissions from housed chicken to both indoor temperature and relative humidity, parallel measurements of the water, UA and TAN content and pH of different litter layers would be helpful to improve future parametrization.

### 4.1.3 The $NH_3$-transfer resistance of chicken houses

The inversion-derived resistance within the chicken houses, R* at NC2B typically ranged from 10000 s m$^{-1}$ up to 50000 s m$^{-1}$ with strong variations. According to Pinder et al. (2004), from a dairy manure storage sub model with parameter tuning, the surface resistances of crust with wheat straw ranged between 0.1 to 0.4 day m$^{-1}$, which corresponds to 8640 s m$^{-1}$ to 34560 s m$^{-1}$. As no obvious correlation between R* and environmental factors was found, it remains unclear that by which parameters





the R* are affected. Based on the conditions of two chicken houses at NC2B, the sensitivity test of using constant R* value to simulate 1-year $NH_3$ emissions suggested that the R* values led to the best agreement with the measurements were 16700 and 14369 s m$^{-1}$, respectively. It is worth noting that the best-fit R* values for each house are smaller than the mean or median values of the inversion derived R* values. This indicates that a relatively small R* value leads to a good approximation of the

fraction of TAN pool being depleted through $NH_3$ emission, while R* becomes less effective on restricting $NH_3$ emissions as its value increases. For the House A, change of R* from 8350 to 33400 s m$^{-1}$ caused the ratio (of simulated to measured $NH_3$ emission) decrease from 1.24 to 0.75. Likewise, changing R* within the House B from 7185 to 28740 led to the ratio ranged between 1.25 to 0.73. The varying of R* value by a factor of 2× resulted the total $NH_3$ emission for a whole year period changing approximately 25 %. This implies that under current housing conditions, the total annual $NH_3$ emission is not strongly

influenced by the resistance within the houses. Instead, resistance plays more crucial role in affecting the short-term emissions. Large resistance limits the emission initially, but leads to the TAN accumulation to allow larger emissions at a later point in time, therefore reducing the overall sensitivity to R* for annual timescales.

### 4.1.4 Implications for the idealised simulations

As shown in Fig. 6 and Fig. 7, it can be seen from dry simulations (i.e., without precipitation) that the annual mean $P_V$ was

relatively small and can drop to approximately zero when temperature is low. It indicates that the UA hydrolysis is hardly to take place. In contrast, the $P_V$ were much higher in hot and wet regimes, reflecting an effective hydrolysis of UA. It is notable that the $P_V$ declines at very high RH levels using the new RH parametrization. This is mainly because the UA hydrolysis is considered to be optimum at 80 % and higher RH, but the TAN concentration becomes lower as the excretion contains more water when the ambient environment is humid, thereby providing a "diluting" effect.

From Fig. 7a, the $P_V$ rate is seen to grow exponentially as a function of temperature for the 20 % RH simulations. It is similar to the impact of temperature on UA hydrolysis and also the Henry's Law relationship. Conversely, for a humid environment with RH at 100 %, there is a smaller increase of $P_V$, showing a logarithmic-like trend. These differences are consistent with different amounts of TAN under the two cases. When there is sufficient TAN produced from the UA hydrolysis, the resistance can become the key limiting factor to emission from the system. Conversely, in low-humidity environments, as the UA

hydrolysis is limited, the produced TAN is readily removed through the atmospheric release of $NH_3$, with total emission limited by the UA hydrolysis rate. Therefore, the rise of temperature under dry conditions provides a larger increase in $NH_3$ emissions.

From Fig. 7b, it is worth noting that the decrease of $P_V$ occurs when the RH slightly exceeds 90 % rather than 80 %. A more obvious sharp decline can be seen from the 15 °C simulations. As discussed, there is a "diluting" effect on the TAN concentration when the RH is over a certain level. The possible reason why this turning point does not occur at the 80 % RH

where is the factor RH reaches the optimum can be summarised as follows. The $P_V$ rates in these simulations represent the integral of a whole year. The "diluting" more water to dissolve TAN at high RH affects the instantaneous emission without changing the amount of TAN pool. Low emissions in the earlier stage can therefore cause a larger emission potential in the later stage due to accumulation of TAN.

The overall implication of these idealized simulations is to demonstrate the close interplay between water availability and

temperature, where temperature always increases volatilization (partitioning in favour of the gas phase), whereas a small amount of water is needed to facilitate UA hydrolysis, increasing $NH_3$ emissions, while excess water availability dilutes the



TAN pool, thereby reducing $NH_3$ emissions. These same principles also apply for emissions from manure application to crops and for backyard birds, where precipitation and run-off become more important.

**4.2 Spatial and temporal variations of $NH_3$ emission**

The $NH_3$ emission from chicken agriculture differs substantially across regions, both because of different chicken number
distributions (Supplementary Fig. S10), as this affects total nitrogen excretion, and because of different volatilization rates, as shown by the $P_V$ values. The largest $NH_3$ emission is calculated for regions between 20 ~ 40 °N, which corresponds to the highest chicken density and associated manure application to land. The animal number and the amount of nitrogen from excretion have a first order effect on the magnitude of emissions. Considering the variations in $P_V$, there is most estimated variation in $NH_3$ volatilization of manure spreading and backyard. The $P_V$ rates of backyard chicken excretion were much
lower in China and Southeast Asia by comparison with manure land application, because the wash off is a major loss of nitrogen pools in these regions, especially during non-cropping periods when chicken manure is not applied to land (according to our model approach), while backyard birds lead to outdoor $NH_3$ emissions all year round (including during non-cropping periods with high precipitation).

It should be noted that from the northern India to Tibet, the $P_V$ rate declines sharply from 40 % to below 6 % from all categories.
This indicates that a sudden change from hot and wet conditions to cold and dry conditions causes the volatilization rate drops dramatically in Tibet compared with India. This example clearly illustrates how the fraction of nitrogen volatilised as $NH_3$ is strongly linked to meteorological and related environmental conditions.

The AMCLIM-Poultry simulations also showed strong seasonal variations of $NH_3$ emissions from manure land spreading and backyard chicken excretion. The seasonal distributions (as illustrated by Fig. 13) were caused by changes in meteorological
conditions, with high $NH_3$ emissions in summer due to the high temperature influencing $NH_3$ emissions from housing and backyard birds. Even larger seasonal differences are seen in the modelled emission estimates for land application of manure, because this combines both the direct effects of environmental variation (temperature and water effect on $P_V$) with seasonal differences in the estimated timing of manure application to land. Paulot et al. (2014) found that maximum $NH_3$ emissions from manure fertilising can occur from April to September depending on the local management. For example, they found that
emission peaks in spring occurred in Europe, while summer emission peaks occurred in part of the US and China. These differences reflect a combination of agricultural timing and the meteorological/environmental drivers (Hertel et al., 2011). Riddick et al. (2016) also showed the maximum emissions usually occur in April-June or July-September. The findings in present study are broadly consistent and demonstrate for the first time on a global scale how emissions from managed poultry (chicken) are dependent on both short-term meteorology and long-term regional climatic differences. Contrary to manure
spreading and backyard birds, the seasonal variations of $NH_3$ emissions from chicken housing were much smaller due to the partly controlled environment and the assumed absence of precipitation/run-off within the houses.

**4.3 Uncertainty and limitations**

**4.3.1 Simulation of emissions from chicken housing and storage**

For simulating $NH_3$ emissions from chicken housing, the largest uncertainties are mainly associated with the model
parameterizations linked to temperature (T) and relative humidity (RH). As all the measurements used were from the US



chicken farms, the modelled values of the RH and T parametrizations (Fig. 3) provide only a first estimate to represent variation in climatic conditions on a global scale.

According to our methodology, the parametrization of Fig. 3 is applied to all housed chicken across different climates. However, it is possible that a substantial number of chicken houses are not climate controlled in any way. For example, in tropical countries intensively managed chicken houses may not have any (or only limited) heating and ventilation systems. In this context, a larger fraction of chicken houses may be naturally ventilated throughout the year because cold days are usually very rare. In this case, the temperature inside the chicken house would be simply 2-5°C above the outdoor temperature due to the heat generated by the chicken themselves, with airflow rates are related to natural wind speed. In such a naturally ventilated situation, there may be no steady state between the $NH_3$ emission from the surface and the removal through ventilation. With the availability of appropriate data, such altered ventilation regimes could easily be included in the AMCLIM-Poultry model, and would be expected to show an even larger temperature dependence for chicken housing emissions than estimated using the present parametrization.

Second, due to lack of other data, the new parametrisation for UA hydrolysis is primarily derived from specific chicken houses, under US conditions. These chicken houses had explicit clean out dates for the dataset, which allows the model to be run under a specific initial condition that each nitrogen pool is empty at the beginning. It remains unclear how the model will perform with the new parameterizations for chicken houses that are already loaded with manure. Meanwhile, the equations given by the previous study of Elliot and Collins (1982) resulted in a large discrepancy between the modelled values and measured data during the earlier stage of the simulations. It is evident that there is a need for further experimental datasets for a wider range of climate conditions, including all available indicators ($NH_3$ emissions data and ventilation data accompanied by both temperature and relative humidity, stocking timing and ideally data on manure characteristics). From a modelling perspective, a possible approach of introducing different vertical layers into chicken litter could be useful to investigate the effect of adding fresh bedding onto old, deep litter. However, the additional complexity would need to be judged against the potential benefits.

Third, as the litter in the houses are not subject to precipitation or evaporation, the water amount of the system is calculated from the excretion mass and the equilibrium moisture depending on the RH and temperature. The model is not able to simulate the evaporation from the litter in the chicken house. Therefore, the litter moisture is assumed to be at equilibrium. The weakness of this method is that the initial water within the excretion is not accounted for, which might cause uncertainty.

Fourth, the indoor resistance for $NH_3$ transfer within the chicken houses (R*) needs further investigation. Pinder et al. (2004) applied indoor resistance with dairy houses that were tuned as a function of temperature. English et al. (1980) developed a series of mass transport coefficients given as a function of wind velocity. As there were no specific correlations between environmental factors and the resistance found in this study, we used a constant value in the simulations rather than parameterised. While this provides a significant uncertainty for short term (e.g., daily) fluctuations in $NH_3$ emissions, model feedback reduces the sensitivity over annual timescales, as slow emission earlier (associated with high R*) allows increased emission at a later stage, and *vice versa*. While measurement approaches to estimate R* would be welcome (e.g., using water vapour loss from wetted surfaces or other tracers), the value of R* is therefore not considered the largest uncertainty in the seasonal and annual simulations.





The version of AMCLIM-Poultry applied here does not explicitly treat NH₃ emission from stored chicken manure as a separate step. Emissions from in-house storage of manure are considered as part of the housing calculations, while losses in the field are linked to conditions for land application of manure. For the purpose of the model, which focuses primarily on assessing the climatic dependence on NH₃ emissions, it is assumed that the climatic dependence of emissions from any storage of chicken

manure outside of animal houses and prior to manure spreading follows the same climatic or intermediate climatic dependence between housing and manure spreading. Future work may consider the case to include an additional AMCLIM module for outdoor storage of chicken manure, where the main uncertainties concern: a) providing a basis to estimate the appropriate outdoor manure storage time according to climate and regional practice, b) providing a basis to consider depth and surface area of stored manure, c) providing a basis to estimate the fraction of manure that is stored outside or under cover. Although

the input assumptions are expected to introduce substantial uncertainty, the actual simulations would represent a straightforward extension of the AMCLIM-Poultry approach.

**4.3.2 Simulation of emissions from agricultural land**

Outdoor NH₃ emission from chicken manure consists of two parts: manure fertiliser from broilers and layers applied for crops and backyard chicken excretions left on land and pastures. A major uncertainty in the simulations is the amount of nitrogen

input from chicken manure to crops. There are multiple management options for chicken manure, including composting, burning, and various storage (FAO, 2018). The amount of nitrogen applied for individual crops as input to the model might be overestimated due to the simple comparison method in this study. Meanwhile, as simulations for both processes were run under natural environments, the following parameterizations incorporated in the model also cause uncertainty.

First, the pH of the substrate can greatly affect the NH₃ volatilization by influencing the UA hydrolysis and the TAN partitions

that determines the surface concentration of gas phase NH₃. The pH of the system is dependent to chicken manure pH and soil pH. The chicken manure pH is mostly alkaline, with reported measurements in a range of 7.23 to 9.1 (Sommer and Hutchings, 2001). For the soil pH, there are spatial variations in the geographical distribution. The typical values depending on the crop types range between 5.8 to 7.0, which is usually lower than the pH of chicken manure (Riddick et al., 2016). A major difficulty in determining the pH of the system is because the hydrolysis of UA and NH₃ production can change the soil pH. The NH₄⁺

produced by the decomposition of UA disassociates to form gaseous NH₃, resulting in H⁺ consumption, resulting a sharp increase of soil pH in the initial period and then decrease again in the following days (Chantigny et al., 2004). Móring et al. (2016) proposed a dynamic scheme for simulating soil pH in a field scale model and had a reasonable approximation against measurement. In a following study (Móring et al., 2017), it suggested that a fixed value for soil pH can be used in the modelling of NH₃ emissions in large scales, but the value is uncertain and can differ across regions. Due to the complexity of determining

precise pH, a constant value of 8.5 characteristic for solid chicken manure (Elliot and Collins, 1982; Riddick et al., 2017) is used for simulations. While the assumption of the high pH value results in more rapid UA hydrolysis and higher surface concentration of gas phase NH₃, leading to more emissions, the present approach was found to agree well with the measured NH₃ emissions for housed chicken and is consistent with the approach validated by Riddick et al. (2018) for wild seabird emissions across different climates..

Second, nitrogen pools including UA and TAN are determined by source and loss, while one of the major loss of nitrogen in land spreading simulations is through run-off. The model used a relatively simple approach to calculate the run-off. A coefficient multiplied by the amount of each N-containing component. The coefficient is a product of two variables, a wash-off factor and the water available for wash-off. The wash-off factor was set to 1 % mm⁻¹ rain for run-off of UA and TAN and





0.5 mm$^{-1}$ rain for run-off of manure based on the study of Blackall (2004). The available water equals to the total amount of water excluding the water absorbed by the manure that is simply assumed to be twice as the excretion. Although similar parameterization has been validated by the site measurements for seabird colonies (Riddick et al., 2017), there is potential to develop more sophisticated approaches that might be better adapted to simulate emissions from chicken globally.

Third, the model estimated the NH$_3$ emission without considering the deposition of NH$_3$ onto the vegetation. Based on previous studies, a large fraction of NH$_3$ emitted from the surface TAN pool is considered to be captured by vegetation, which could reach 75 % in the case of outdoor bird excreta under a vegetation canopy (Riddick et al., 2016). From the Bouwman et al. (1997) study, plant recapture of NH$_3$ was estimated to vary from 0.8 in tropical rainforests to 0.5 in other forests to 0.2 for other vegetation. Riddick (2012) estimated the overall capture fraction at 59 % on soil and 73 % on vegetation from seabird -

derived nitrogen experiments, taking account of different seabird habits. However, the capture of NH$_3$ on vegetation is poorly constrained and is dependent to canopy features and boundary layer meteorology (Sutton et al., 2013). Because chicken manure is mostly applied to bare fields, there is not much vegetation capture of NH$_3$ at the earlier stage, therefore this effect is not included in the present study. However, such an effect can be relevant for free-range chicken that are kept outdoors under a woodland canopy (Bealey et al., 2014), so this effect would warrant further consideration if such practices became widespread.

Fourth, in addition to the atmospheric NH$_3$ emission, canopy recapture of NH$_3$ and the runoff, there are other processes influencing the nitrogen pathways, such as losses through nitrification and denitrification, that are not currently included in the AMCLIM-Poultry model. Nitrification is in general an aerobic process which is mainly influenced by the oxygen availability in the soils, with other controls on it including soil water content and soil temperature, while denitrification is generally an anaerobic process, dependent on soil porosity, soil water content, temperature and some other empirical coefficients

(Butterbach-Bahl et al., 2011). As the major objective of this study is to quantify the NH$_3$ emissions from practice relevant to chicken farming, these pathways have not been included.

**4.4. Potential to consider NH$_3$ mitigation scenarios.**

The process-based approach of the AMCLIM-Poultry model lends itself well to the opportunity to assess the implementation of possible management options to abate NH$_3$ emissions. Of the many measures for reducing NH$_3$ emissions as described by

the UNECE (Bittman et al., 2014) several of them could be incorporated as part of future model development, e.g.:

  a)  Measures to optimize animal diets, reducing excretion per animal. Such measures could be incorporated in the estimated amount of excretion per bird.
  b)  Measures to reduce moisture in poultry houses, to reduce UA hydrolysis. Such measures could be incorporated into the relationship between indoor and outdoor conditions for relative humidity.

c)  Measures to reduce temperature of stored manure, to reduce UA hydrolysis and NH$_3$ emission. Such measures could be included in a possible future AMCLIM module on manure storage, by altering model temperature.
  d)  Measures to alter the timing of manure application to favour land application under cool conditions. This could be included by altering assumed ambient temperature compared with seasonal averages.
  e)  Measures to incorporate poultry manure immediately into the soil. This could be included empirically based on

35       alteration of atmospheric transfer resistances, or by more detailed development of several vertical layers or the model nitrogen pools (cf. Riedo et al., 2002).

...





While such considerations represent opportunities for future work, they highlight how a the AMCLIM-Poultry model is well suited to consideration of NH$_3$ emissions abatement scenarios.

**5 Conclusions**

This paper presented the simulated NH$_3$ emission from global chicken farming by using the AMCLIM-Poultry model,
including consideration of meteorological effects and simplified agricultural practices. The AMCLIM-Poultry model was designed based on underlying physics and chemistry, supported by evidence from experimental studies.

The magnitude of total NH$_3$ emissions from chicken farming estimated by the AMCLIM-Poultry based on 2010 was 5482.3 Gg N yr$^{-1}$, which accounts for approximately 13 % of agriculture-derived NH$_3$ emissions (Crippa et al., 2016). High NH$_3$ emissions were from South and East Asia, Europe and southeast US. These regions also had high NH$_3$ volatilization rates,
expressed as the percentage of excreted nitrogen (P$_V$) that is volatilized as NH$_3$. The tropics often had high P$_V$ values being up to five times than cold or dry regions, which illustrates how large NH$_3$ emission potentials are expected under hot and wet conditions. Agricultural activities related to chicken represent appreciable NH$_3$ sources, indicating that currently increasing NH$_3$ emissions accompanied by increasing chicken density (FAO, 2018) is important, especially as climate change is also expected to increase NH$_3$ emissions, as demonstrated by the spatial comparisons of the model.

Based on 2010, the model estimated that 24.2 % of the total excreted nitrogen was volatilized as NH$_3$ emission from chicken housing. The total NH$_3$ emission was 2185.5 Gg N, where 1374.7 Gg N was from broilers and 810.8 Gg N was from layers. For the land based emissions, global NH$_3$ emissions were 2582.3 Gg N from manure fertiliser applications for crops and 714.5 Gg N from backyard chicken excretion, respectively, with strong spatial and temporal variations. In the current model approach, NH$_3$ emissions from manure storage are incorporated as 'in-house' storage with housing emissions. Further
information on variation in practices is needed as a basis to estimate NH$_3$ emission from out-door storage of chicken manure, although the overall climate effect is expected to be midway between that for housing (covered outdoor storage) and land-spreading (uncovered storage).

Contrary to empirical approaches, this study uses a process-based method to quantify NH$_3$ emission from chicken, which provides a foundation for estimating emissions from other livestock types, based on theoretical considerations. The calculation
of P$_V$ values is an asset of the model, which provides an insight of how environmental interactions will affect the NH$_3$ emissions, and which could also be applied to consider scenarios using emission abatement options. Strong spatial variation of P$_V$ implies that a single empirically derived emission factor would not usually reflect reality under different climate conditions. The results of this study show increased emissions under warm conditions, pointing to an expectation that climate change will increase chicken NH$_3$ emissions globally. The different relationships for housed chicken (primarily temperature
and humidity dependence) and for backyard birds and manure spreading (primarily temperature and precipitation dependence), indicate that the net effect of climate change on regional emissions will depend on the relative composition of chicken types and management.





**Data availability**

Model results presented in this study are in netCDF format and can be freely accessed (embargoed) through Edinburgh DataShare (https://datashare.is.ed.ac.uk/handle/10283/3644, Jiang et al., 2020).

**Author contribution**

JJ, DS and MAS designed the research. JJ developed the model code and performed the simulations. AU and GT prepared the model input data. JJ, DS and MS analysed the model outputs and wrote the paper. All authors contributed to the interpretation of results and critical revision.

**Competing interest**

The authors declare they have no conflict of interest.

**Acknowledgements**

JJ gratefully acknowledges Dr Stuart Riddick for providing the script of the GUANO model, the GLEAM group from the FAO for preparing global livestock data and support from University of Edinburgh and UK Centre for Ecology & Hydrology (CEH). The project received funding from the NEWS UK-India funded by the UK Biotechnological and Biological Research Council (BBSRC) and the UK Natural Environment Research Council (NERC). MAS is grateful for support from the Global
Environment Facility (GEF) through the UN Environment Programme (UNEP) for the project "Towards the International Nitrogen Management System" (INMS), and from the UKRI under its Global Challenges Research Fund for support of the GCRF South Asian Nitrogen Hub NE/S009019/1, and from NERC for National Capability support, including through the CEH SUNRISE project.  We thank US-EPA for providing public access to the AFO datasets and Steen Gyldenkaerne for advice at an early stage of the project.



**Appendix**

| Abbreviation | Unit | Model Variable |
|---|---|---|
| $f_N$ | g N g excretion$^{-1}$ | N content of chicken excretion |
| $f_{UA}$ | | Fraction of uric acid in chicken excretion |
| $F_e$ | g N m$^{-2}$ s$^{-1}$ | Total nitrogen excretion rate from chicken |
| $F_{H2O}$ (evap) | g m$^{-2}$ s$^{-1}$ | Evapouration |
| $F_{H2O}$ (rain) | g m$^{-2}$ s$^{-1}$ | Precipitation |
| $F_{NH3}$ | g N m$^{-2}$ s$^{-1}$ | Net rate of conversion of TAN to gaseous $NH_3$ within litter/manure |
| $F_{removal}$ | g N s$^{-1}$ | Removal of $NH_3$ through ventilation in the chicken house |
| $F_{surface}$ | g N s$^{-1}$ | Total flux of $NH_3$ from surface litter in the chicken house |
| $F_{TAN}$ | g N m$^{-2}$ s$^{-1}$ | Conversion rate of uric acid to TAN |
| $K(T,pH,RH)$ | s$^{-1}$ | Function of temperauture, pH and RH influencing uric acid hydrolysis rate |
| $k_{pH}$ | | Function of pH influencing uric acid hydrolysis rate |
| $k_{RH}$ | | Function of RH influencing uric acid hydrolysis rate |
| $k_T$ | | Function of temperauture influencing uric acid hydrolysis rate |
| $m_E$ | | Equilibrium moisture content of litter/manure |
| $M_{available\ water}$ | g m$^{-2}$ | Mass of water in the system that is available for washoff |
| $M_{excretion}$ | g m$^{-2}$ | Mass of excretion |
| $M_{H2O}$ | g m$^{-2}$ | Mass of water in the system |
| $M_{H2O}$ (e) | g m$^{-2}$ | Mass of water in the excretion |
| $M_N$ | g N m$^{-2}$ | Mass of nitrogen components |
| $M_{N-runoff}$ | g N m$^{-2}$ | Mass of instant runoff for nitrogen components |
| $M_{TAN}$ | g N m$^{-2}$ | Mass of nitrogen in form of TAN |
| $M_{UA}$ | g N m$^{-2}$ | Mass of nitrogen in form of uric acid |
| $N_{Crop}$ | g N m$^{-2}$ | Amount of total N application for individual crops |
| $N_{Crop\_Chicken}$ | g N m$^{-2}$ | Amount of chicken manure-N application for individual crops |
| $N_{Soil\_Chicken}$ | g N m$^{-2}$ | Amount of available chicken manure-N |
| $N_{Total\_manure}$ | g N m$^{-2}$ | Amount of total N application for all crops |
| $pH$ | | pH of litter/manure |
| $Q$ | m$^3$ s$^{-1}$ | Ventilation rate in chicken house |
| $Q_{available\ water}$ | mm | Pools of water in the system that is available for washoff |
| $r_N$ | mm$^{-1}$ | Washoff factor |
| $R_{runoff}$ | | Runoff coefficient |
| $R^*$ | s m$^{-1}$ | Overall indoor resistance in chicken house |
| $R_a$ | s m$^{-1}$ | Aerodynamic resistance |
| $R_b$ | s m$^{-1}$ | Boundary layer resistance |
| $RH$ | % | Relative humidity |
| $S$ | m$^2$ | Surface area of chicken house |
| $T$ | K | Ground temperature |
| $V$ | m$^3$ | Volume of chicken house |
| $V_{H2O}$ | ml m$^{-2}$ | Volume of water in the manure |
| $z$ | m | Reference height |
| $\chi_{in}$ | g m$^{-3}$ | Air concentrantion of $NH_3$ in chicken house |
| $\chi_{out}$ | g m$^{-3}$ | Air concentrantion of $NH_3$ of embient environment |
| $\chi_{surface}$ | g m$^{-3}$ | Concentrantion of $NH_3$ in litter/manure on the surface |





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





**Graphs**

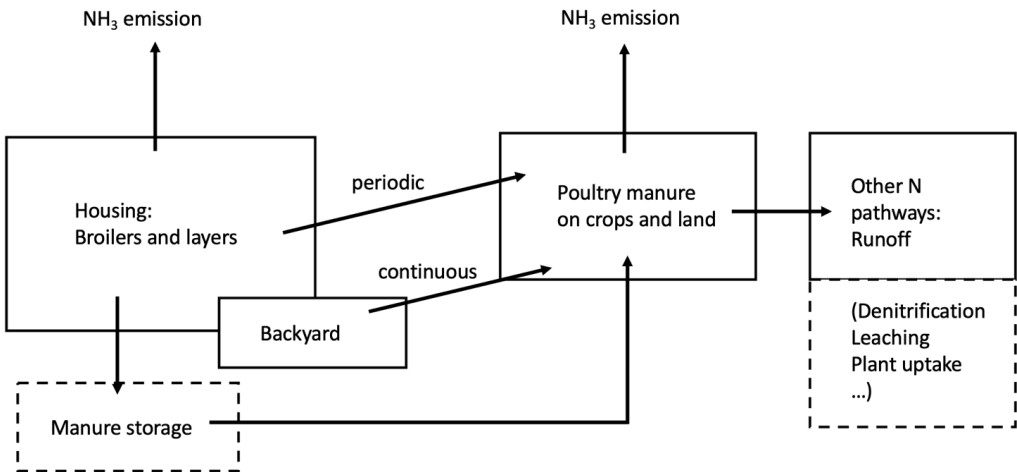

**Figure 1** Schematic of the AMCLIM-Poultry model for estimating NH₃ emissions from global chicken farming following nitrogen pathways from chicken farms to land spreading. Aspects noted in dashed boxes are not investigated in this study.

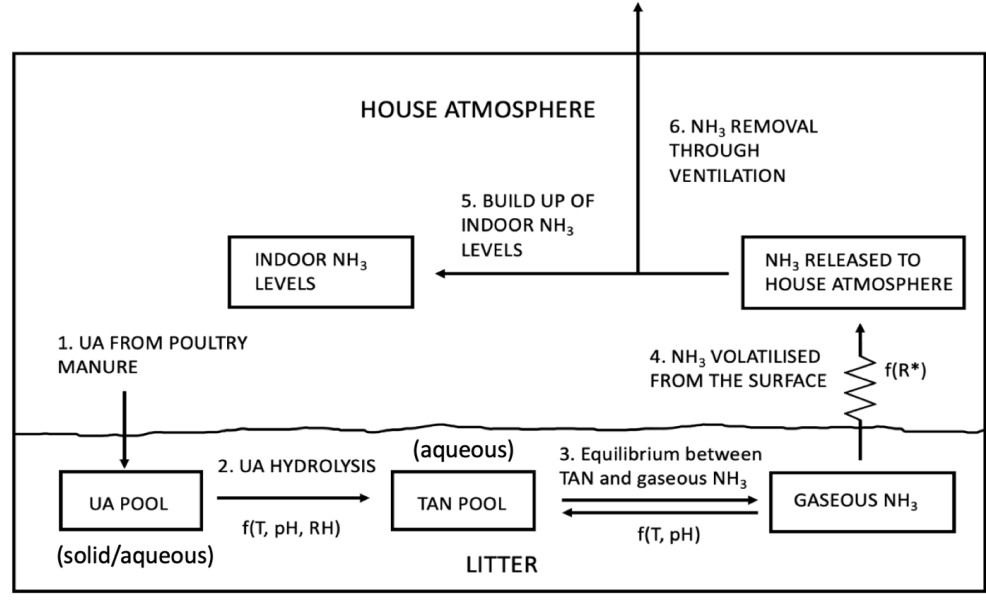

**Figure 2** Schematic of NH₃ volatilization in the poultry house. UA is uric acid; TAN is total ammoniacal nitrogen, R* is the resistance for gaseous transfer from the litter surface to the in-house atmosphere (adapted from Elliott and Collins, 1982)





**Figure 3** Factors affecting UA hydrolysis rate in chicken houses. Red curves represent the results from Elliott & Collins, 1982. Blue curves represent results from this study using data from the 2012 Monitored AFOs (see Sect. 2.2.1). a) Influence of temperature on UA hydrolysis. b) Influence of relative humidity on UA hydrolysis at optimum temperature condition (≥35 °C). Dashed line is the extrapolation of factor RH as a function of RH due to lack of data when relative humidity was below 40 % in the AFO experiments.



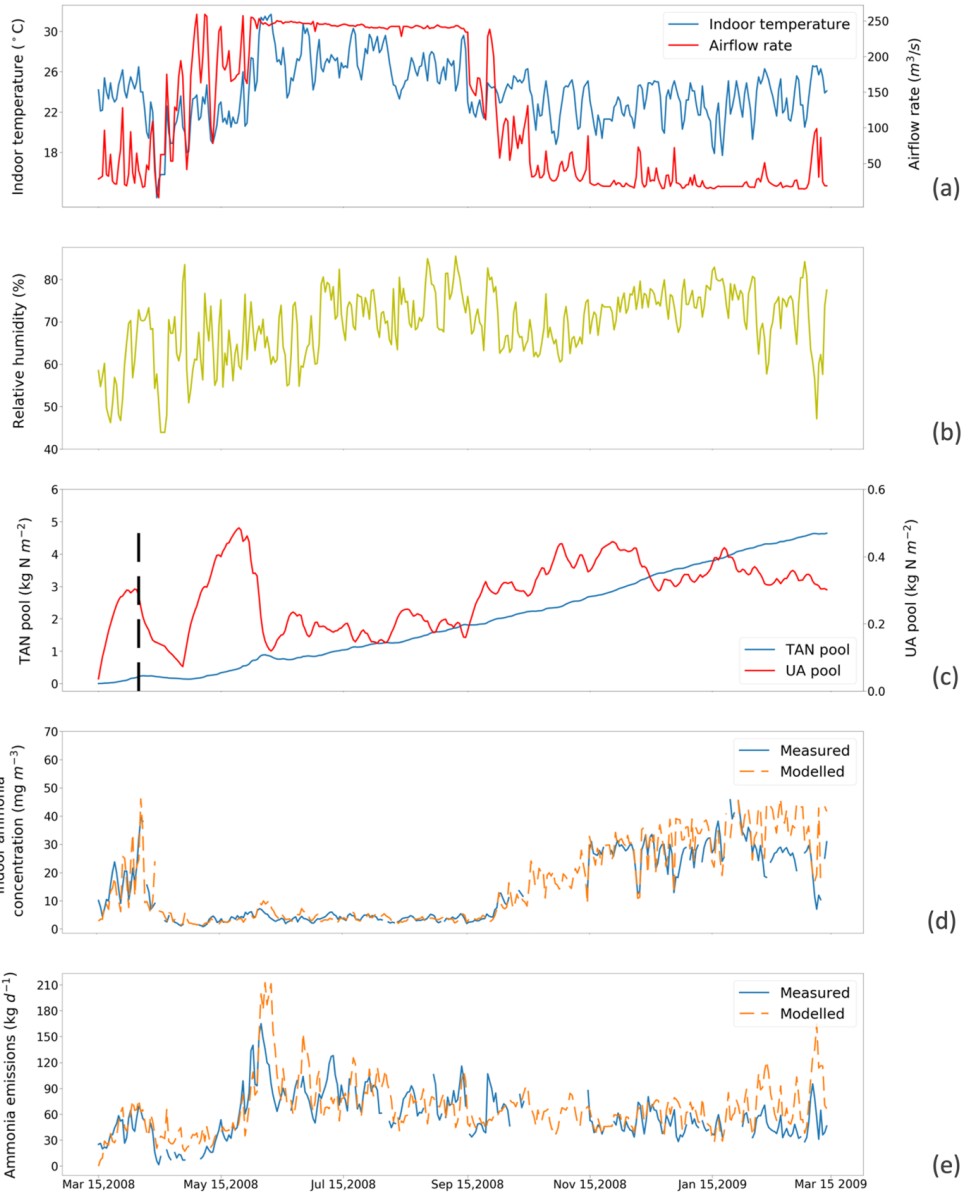

**Figure 4** Site simulations using fixed resistance (R*) value of 16700 s m$^{-1}$ for House A at site NC2B, Nash, North Carolina from March 15 to March 15, 2009. a) Measured daily mean indoor temperature and airflow rate of the house. b) Measured daily mean relative humidity of the house. c) Modelled TAN pool and UA pool. The black dashed line indicates the house emptying date of April/09/2008. d) Comparison between measured and modelled indoor NH$_3$ concentrations of the house. e) Comparison between modelled NH$_3$ emissions and calculated NH$_3$ emissions from measured indoor concentrations. The simulation illustrated uses the new parametrization (based on the AFO data, Fig. 3) for relative humidity dependence of UA hydrolysis.







**Figure 5** Site simulations using fixed resistance (R*) value of 14369 s m$^{-1}$ for House B at site NC2B, Nash, North Carolina from March 15 to March 15, 2009. a) Measured daily mean indoor temperature and airflow rate of the house. b) Measured daily mean relative humidity of the house. c) Modelled TAN pool and UA pool. The black dashed line indicates the house

5 emptying date of June/03/2008. d) Comparison between measured and modelled indoor NH$_3$ concentrations of the house. e) Comparison between modelled NH$_3$ emissions and calculated NH$_3$ emissions from measured indoor concentrations. The simulation illustrated uses the new parametrization (based on the AFO data, Fig. 3) for relative humidity dependence of UA hydrolysis.



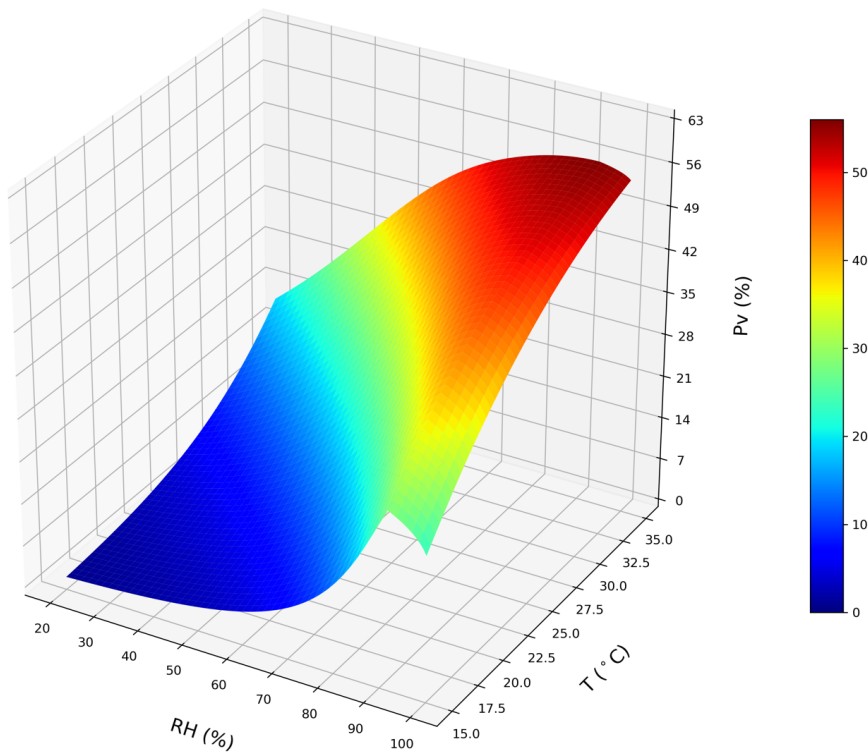

**Figure 6** A conceptual 3-D sketch of NH₃ volatilization rate (Pᵥ (%)) that is driven by temperature (T) and relative humidity (RH) The surface plot is derived from a set of idealised steady state simulations with zero precipitation to simulate dependences for emissions from chicken housing  (see Sect. 3.1.2 Shown using the new parametrizations for T and RH).



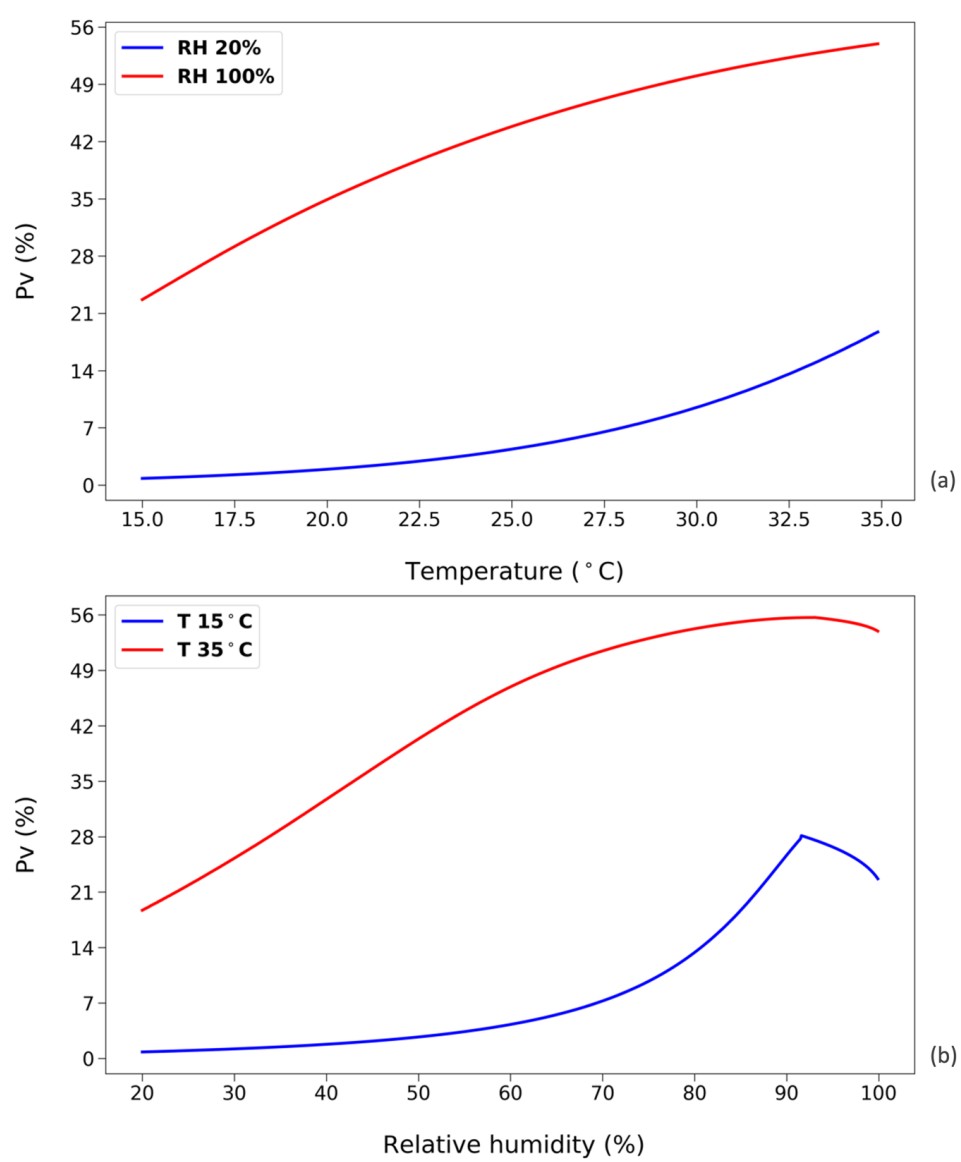

**Figure 7** Curves that represent 4 different regimes from Fig. 6. a) The NH₃ volatilization rate (Pᵥ (%)) under dry (20 % relative humidity, RH) and wet (100 % RH) conditions, respectively. b) The NH₃ volatilization rate (Pᵥ (%)) under 15 °C and 35 °C, respectively. (See Sect. 3.1.2, shown using the new parametrizations for temperature and RH).



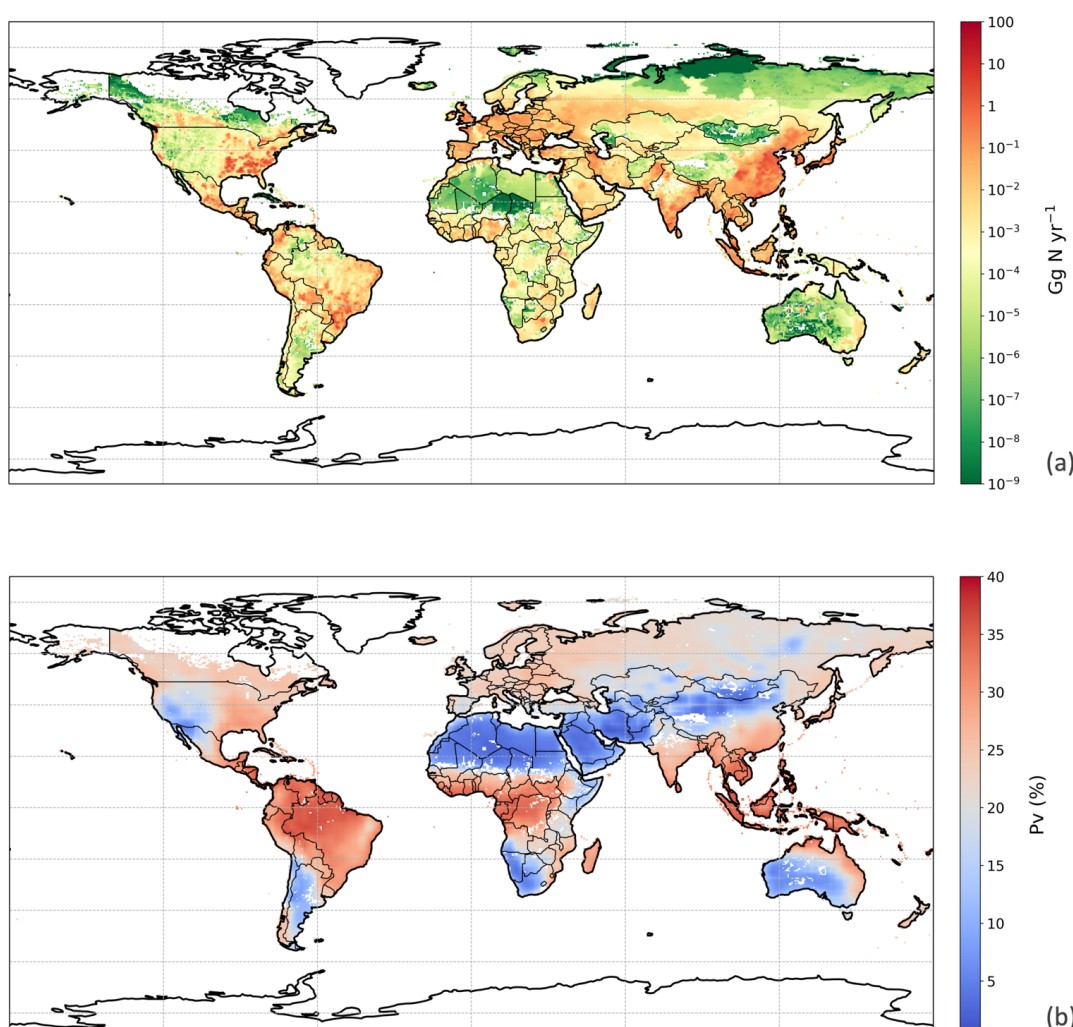

**Figure 8** Simulated a) annual global NH$_3$ emissions (Gg N yr$^{-1}$) from chicken housing in 2010. b) Percentage of excreted nitrogen that volatilizes (P$_V$, %) as NH$_3$ from chicken housing in 2010. The resolution is 0.5°×0.5°. For the simulation shown the RH parametrization for UA hydrolysis is taken from Elliott and Collins (1984). Figure S9 shows the results of using the

5   RH parametrization based on new parameterization from AFOs monitored data, for comparison.



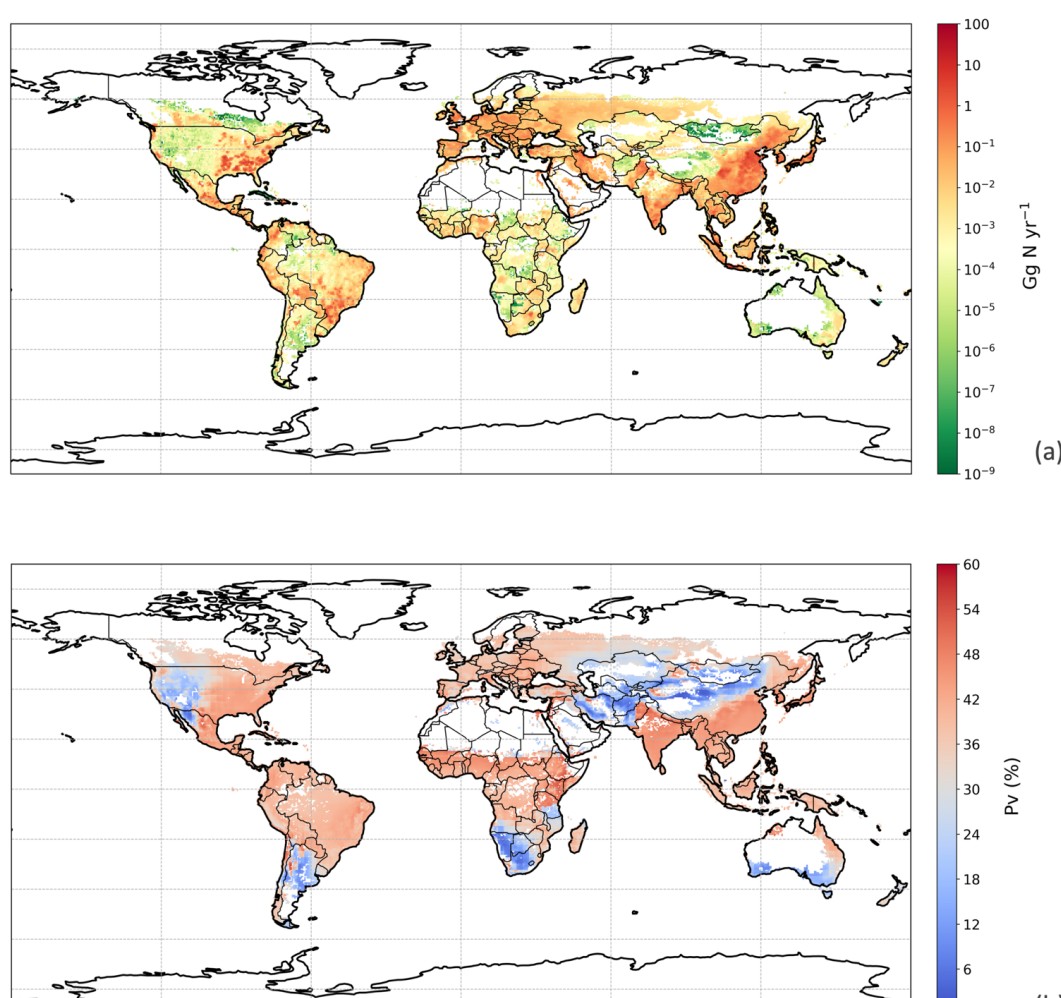

**Figure 9** Simulated a) annual global NH₃ emissions (Gg N yr⁻¹) from chicken manure application for crops in 2010. b) Percentage of excreted nitrogen that volatilizes (Pᵥ, %) as NH₃ from chicken manure application for crops in 2010. The resolution is 0.5°×0.5°.



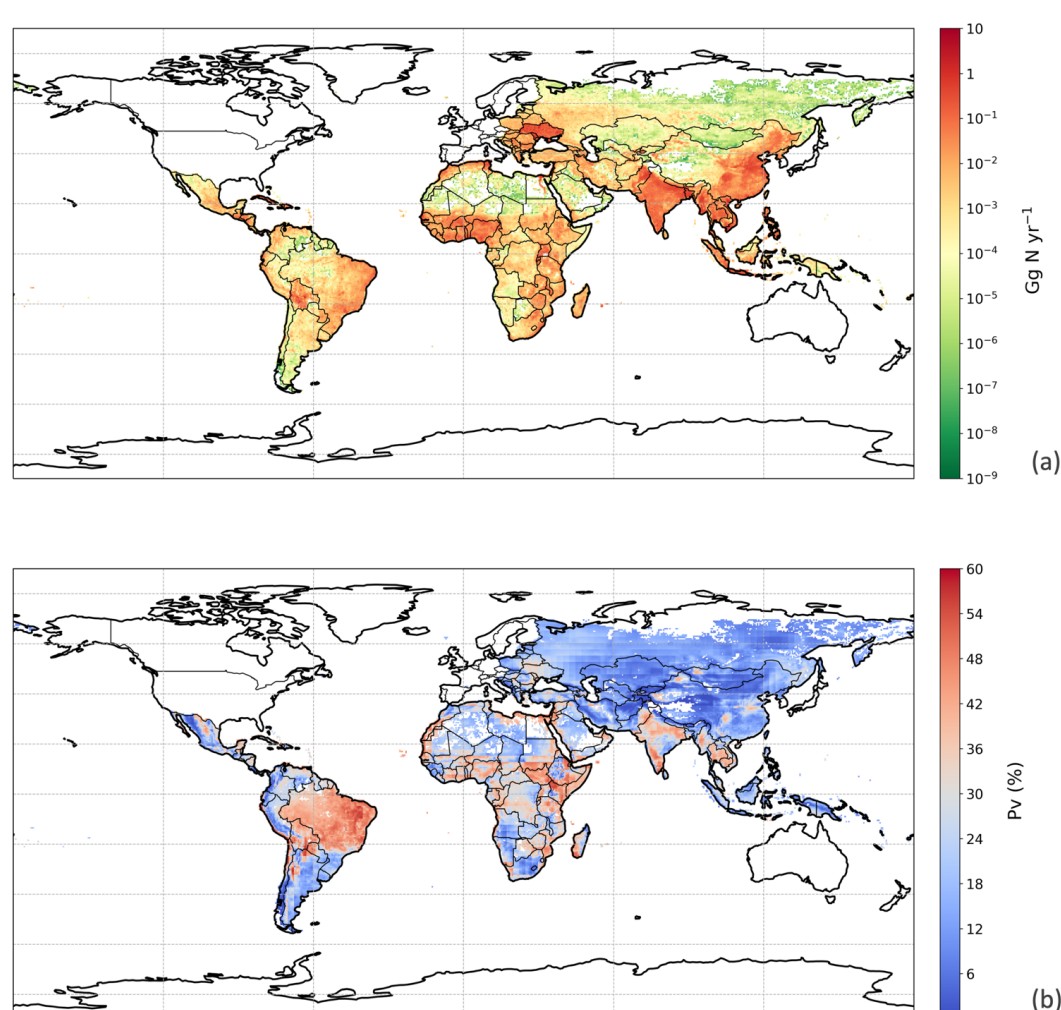

**Figure 10** Simulated a) annual global NH₃ emissions (Gg N yr⁻¹) from backyard chicken in 2010. b) Percentage of excreted nitrogen that volatilizes (P$_V$, %) as NH₃ from backyard chicken in 2010. The resolution is 0.5°×0.5°.



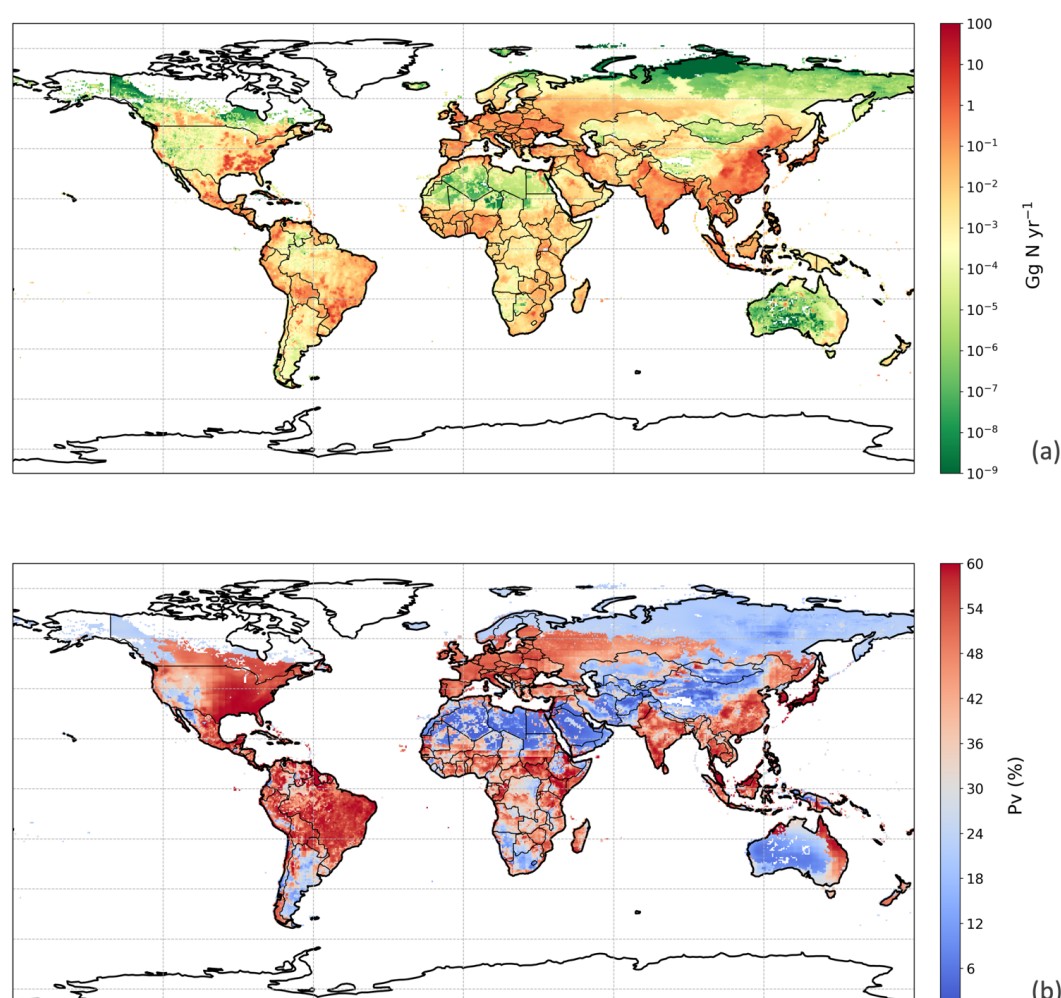

**Figure 11** Simulated a) annual global NH₃ emissions (Gg N yr⁻¹) from chicken agriculture in 2010. b) Percentage of excreted nitrogen that volatilizes (P$_V$, %) as NH₃ from chicken agriculture in 2010. The resolution is 0.5°×0.5°.





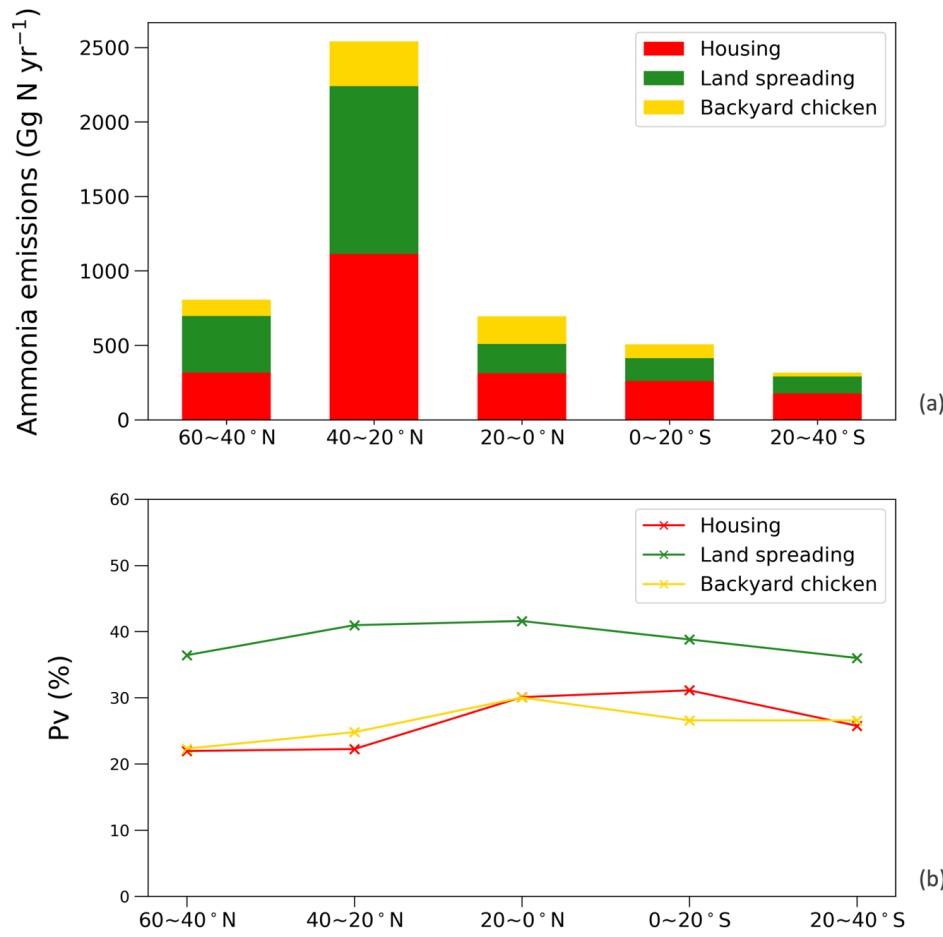

**Figure 12** Simulations for chicken housing, manure applications to crops and land spreading of backyard chicken manure in 2010 given in regions. a) annual global $NH_3$ emissions (Gg N $yr^{-1}$). b) Percentage of excreted nitrogen that volatilizes ($P_V$, %) as $NH_3$.





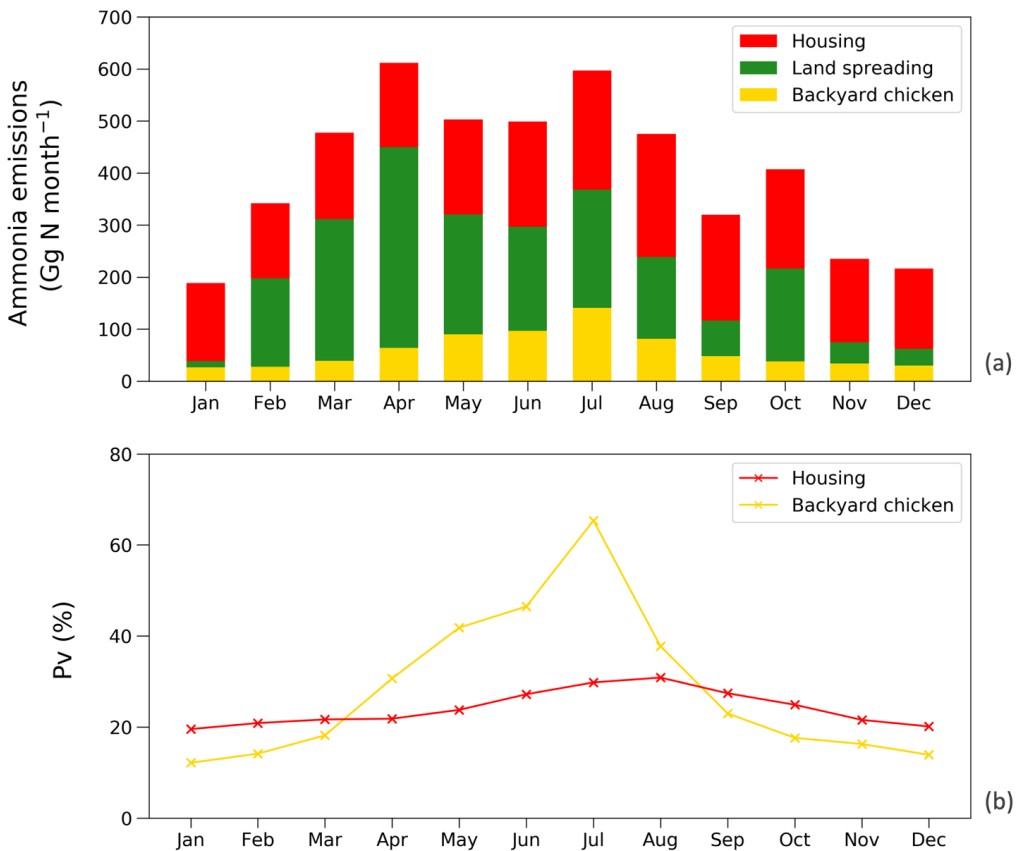

**Figure 13** a) Monthly NH$_3$ emissions (Gg N yr$^{-1}$) from chicken housing, manure applications to crops and land spreading of backyard chicken manure in 2010. b) Percentage of excreted nitrogen that volatilizes (P$_V$, %) as NH$_3$ monthly for chicken housing and land spreading of backyard chicken manure.



**Table 1** Excreted nitrogen from housed and backyard chicken, and estimated NH$_3$ emissions from each practice based at 2010.

| Production system | Excreted nitrogen (Gg N) | Practice | Emission (Gg N) |
|---|---|---|---|
| Broiler and layer | 9017.1 | Housing | 2185.5 |
| | | Land spreading | 2582.3 |
| Backyard chicken | 2178.3 | Left on land | 714.5 |
| Total | 11195.4 | | 5482.3 |

