# Peer review of "A climate-dependent global model of ammonia emissions from chicken farming"

_Biogeosciences, 2020_

## Referee Comment (RC1) · Anonymous Referee #1 · 27 Jul 2020

The paper presents an extensive and well detailed model for NH3 emissions from chickens. As mentioned in the manuscript, most of the current global emission inventories for livestock are based on emission factors without any consideration for regional climate conditions and farming practices. The introduction of a few regional dependent parameters should greatly improve the spatial and temporal variations in the NH3 emissions which is of great value in for example in air quality modelling. The manuscript is well written, structured and easy to read.

Major comments

1. Hourly and Daily timescales; the authors describe in section 2.1 that their model operates at an hourly timescale for outdoors emissions while only at a daily level for indoor emissions. While variations in temperatures inside can reasonably be expected to

be small, emissions will show some variations as a function of the inside temperature, which will lead to variations in the emissions to the outside. If possible add a sentence about the choice for two different timescales and the potential impact.

2. Section 3.1.3; line 17. I would argue that the model does an average job at capturing the overall level of emissions but that some of the major changes at the start of the measurement period seem to be over and underestimated by up to a factor 2 (figure 5 May-June and ∼September). Add some discussion on the main cause for the discrepancy between the modeled and measured emissions.

3. Uncertainties; the various uncertainties within the model are discussed to great extent but what is missing is a final summary and overall estimate of uncertainty. If possible add a table summarizing the various errors and uncertainties, including an expected (back of the envelope) range of uncertainty for each individual error. Similarly add a summary/discussion on the total expected uncertainty, and a summary for the uncertainties in the spatial and temporal distributions (similar to the ranges, at a back of the envelope level).

4. Current inventories; that brings us to a comparison to current inventories which is as of yet missing in the manuscript. Most regional/country scale inventories, to some extent, do have emission totals for chicken housing/open-range chickens. How do the emissions reported in this manuscript compare to some of those emission inventories (for example, UK, Netherlands, Denmark, US, German inventories. . .etc), and did the added complexity of the model improve the overall uncertainty in the emission totals?

5. Similarly, add some discussion on the average Volatilization levels reported in this study compared to those in current literature.

Minor edits and remarks

a. Figure S1, is there any reasoning behind the choice of a third order polynomial?

b. Page 6., line 9, add "of" between lack and knowledge.

---

## Referee Comment (RC2) · Anonymous Referee #2 · 24 Aug 2020

**General comments**

The manuscript "A climate-dependent global model of ammonia emissions from chicken farming" from Jize Jiang et al., describes a model of ammonia volatilization from chicken farming: AMCLIM-Poultry.

The model is based on a simple approach were urea hydrolysis to ammonium and ammonia is implemented for emissions in buildings, in field applied with chicken manure and in farm backyards. A resistance approach is used and specific resistance parameterisation is used for buildings. A simple mass balance approach is used to treat manure water content.

The model is compared to measurements in a few US farms and applied to evaluate worldwide emissions from chicken farming, based on FAO statistics.

The issue is of great interest for the scientific community as ammonia emission is a key component of air quality prediction and environmental impacts and emissions from chicken farming is still not well developed. The presented study is based on the work of Elliot and Collins (1982) for hydrolysis and combined with a resistance approach. The application of the model at the global scale is of great interest, and especially the analysis of the humidity and temperature dependent NH3 emissions as well as the dataset constructed for that purpose.

This manuscript should be published provided some the authors answer some comments on the model design.

- Model: The model is key in this manuscript and it is both very simple but it accounts for the most important processes about the environmental conditions, which makes it effectively very useful. The presentation of the model may however be improved by first exposing clearly, right at the beginning, the hypothesis behind it, second condensing the description in the material and methods only, whereas it is now split between sections, and third, better explicating the model for manure spreading in the field.
    - Regarding model hypothesis, I found several hypotheses that were not always explicit: i)there is no transfer resistance in the litter itself (eq. 7); ii) ammonium is considered the only form of TAN in the liquid phase (eq. 6); iii) the pH is considered not influenced by the UA hydrolysis; iv) $NH4^+$ is consider to be completely free in the litter and soil and not to be bound to soil or litter particles; v) the system is considered to be litter only but no soil; vi) No exports are in the equations but the model is initialised at each house cleaning; vii) there no litter evaporation is considered in the houses, rather an equilibrium is considered.
    - Regarding the description of the model, it would be much easier to read if the whole model could be defined at once in the material and methods: factors affecting UA hydrolysis should be presented in the material and methods. Watch out that the TAN is sum of NH3l+ NH4 and you should justify NH4 >> NH3.
    - Regarding the manure spreading, it is unclear how VH2O is calculated in this situation, and the description of run off is quite unclear.
- UA hydrolysis fitting to RH and TA: Did you try fitting on vapour pressure pvap = RH/100*psat(Ta) ? In addition, did you try fitting on both Ta and RH together?
- Literature: I feel that some important papers may be lacking. In particular, on ammonia emissions data and models from land spreading manure or urea hydrolysis. The literature is much more abundant on dairy cow or pig manure, but I was wondering if and why it would

not be possible to refer to these when building up the model for chicken manure. Some examples given here

- o Ammonia Volatilization after Surface Application of Laying-Hen and Broiler-Chicken Manures. By: Miola, Ezequiel C. C.; Rochette, Philippe; Chantigny, Martin H.; et al. JOURNAL OF ENVIRONMENTAL QUALITY Volume: 43 Issue: 6 Pages: 1864-1872 Published: NOV-DEC 2014. Typos: please check thoroughly the text for typos.
- o The molecular processes of urea hydrolysis in relation to ammonia emissions from agriculture By: Sigurdarson, Jens Jakob; Svane, Simon; Karring, Henrik. REVIEWS IN ENVIRONMENTAL SCIENCE AND BIO-TECHNOLOGY Volume: 17 Issue: 2 Pages: 241-258.
- o Modeling and measurements of ammonia from poultry operations: Their emissions, transport, and deposition in the Chesapeake Bay By: Baker, Jordan; Battye, William H.; Robarge, Wayne; et al. SCIENCE OF THE TOTAL ENVIRONMENT Volume: 706 Article Number: 135290 Published: MAR 1 2020
- o Semi-empirical process-based models for ammonia emissions from beef, swine, and poultry operations in the United States By: McQuilling, Alyssa M.; Adams, Peter J. ATMOSPHERIC ENVIRONMENT Volume: 120 Pages: 127-136 Published: NOV 2015

- Consider shortening the discussion. I found the discussion a bit long with a few redundancy and repetitions.
- A comparison with existing emission factors would be very interesting
- Typos and English. I suggest double-checking the spelling and phrasing of the manuscript.

**Detailed comments**

P2.L17-18: Could you be more specific on which parameters were tested?

P3.Eqns (1-3): In these two equations, the export flux of excretion by removal during house cleaning is not considered. It would be clearer to add it. This would allow all Mexctretion, MUA and MTAN to get down to zero when the house is cleaned.

P4.L1: FTAN is not a conversion rate but a flux. Please consider revising.

P4.L11: and eq. 4: it would be good to give expression of K here rather than in the results section.

P4eq. 6 is not strictly speaking true since MTAN = MNH4+ + MNH3. Does this mean you consider MNH3 negligible compared to MNH4+? You could easily express MNH3 as a function of MNH4+ based on the dissociation constant and pH and then get a corrected expression for equation 6 that accounts for the pH.

P4L26-27: the justification of using the same approach for backyard and field may be more developed. Especially, how the interaction with the soil is treated.

P5L7-8: NH3 is removed but also fresh air dilutes NH3 in the building: both process occur.

P5 eqns 8 and 9: From what I understand here, the litter (or excretions) has a humidity, which is in equilibrium with atmospheric humidity in the building (express by RH and T). This is similar to soil surface humidity that is in equilibrium with the atmosphere just above. Could-you explain the process behind equation 9?

P6L1: The pH should be influenced by UREA hydrolysis, isn't? Could you better justify the choice of fixing the pH?

P6L28: I suggest explicitly stating that Qxout has been neglected.

P6 eq 12-13: fundamentally, this equation would also hold for water in buildings: hence, humidity in the building may depend on the rate of air renewal and the surface humidity. This would mean that pvapin = f(pvapout, Q, R*, pvapsurface) but also that there would a removal flux for humidity also. Proportional to Q*(pvapin-pvapout). Could you elaborate on that and justify better, why evaporation from building is neglected?

P7L3-4: I suggest defining clearly, what the "system" is: is it the litter only, or the litter plus a certain depth of soil?

P7L8-9: Could you explain better why the water amount in the system could not be less than that in the excretion? Indeed, since evaporation occur, the water amount may become lower.

P7, section 2.3: The field application is unclear and would need further details: 1) TAN in soil is known to be in equilibrium with clay, explain why this process is neglected. 2) The evaporation equations as well as the expressions of the resistances are not given and should be detailed, in the supplementary at least. 3) How is VH2O calculated in that situation?

P8L28: but evaporation ay also occur in the building. Please comment.

P8L30: "houses were empty in different months". Please rephrase as this is unclear what it means.

P9eq 18: I suggest using the term Navailable instead of Nsoil_poultry. It is also unclear from the text, whether N_total includes manure and mineral nitrogen

P10L21-22: It is unclear when the building temperature is not used, what temperature is then used? Please clarify.

P10-P11: section 3.1.2 should be in the material and methods section and not in the results as it is a model description to me.

P11 eq 23: To me it would be more logical if urea hydrolysis would be dependent on the excretion humidity %me rather than RH. However, the two are linked. Could you comment on that?

P11-L16-17: "emissions were due to unavailable measurements": this sounds weird: could you rephrase?

P12 section 3.3: the model for manure spreading was not tested at all, while the model for housing was tested. Would there be any dataset to demonstrate the quality of the model for outdoor application? Alternatively, would there be any paper to refer to on that?

P14- L6-13: it is actually unclear in the previous part if the papers how RH and Ta are modelled in houses.

P14-L14-20: I would suggest adding a table with durations, temperatures and may be RH conditions for the different chicken houses managements discussed

P14-15 section 4.1: it is a bit confusing here to understand how the RH-dependency of urea hydrolysis is used in outdoor conditions. Please detail.

P16L1-10: the whole paragraph except last sentence is quite unclear. Please rephrase. In the last sentence, it may not be true that sensitivity is negligible though, since R* may be very variable among situations.

P16-L27-33: Could we not say that for very large RH, since UA hydrolysis is so effective, there is a limiting effect due to the non-availability of total nitrogen in the system after a certain time?

P17L8-9: Difficult to understand. Please rephrase this sentence

P18L26: It is unclear why initial water in excretion is not accounted for. Please rephrase.

P18-last paragraph: this section would need sensitivity tests to better demonstrate that R* does not represent a great uncertainty.

P19 section 4.3.2: In this section a sensitivity to pH would be interesting to show to illustrate the possible effect of changing the manure pH by +- 1 point.

**FIGURES AND TABLES**

Fig 1: explain meaning of arrows

Fig 2: I would suggest adding flows in and out of the farm. In addition, an arrow for dilution through ventilation pointing towards INDOOR NH3 LEVELS may be considered. Watch out that the volatilisation flux is bi-directional. An arrow downwards should be shown.

Fig 3: It is unclear how the UA factors were calculated. 3a: could you give a hint on the significance of the difference between the two curves?

Fig 4d and 5d: I would suggest showing also on the same graph the ammonia concentration at z0 (the compensation point) as it would give ground to better understand the NH3 emissions dynamics.

Fig7: please explicit the fact that the curves are evaluated for yearly datasets. I suggest showing also total UAN remaining before cleaning to show any N-limiting effect on Pv. I also suggest rephrasing: 'NH3 volatilization rate Pv(%) for 4 different RH and Ta regimes….'

Table 1: I would suggest adding percentage of N loss for each production system. In addition, you may consider getting rid of unneeded precision in emission numbers.

---

## Author Comment (AC1) · 30 Sep 2020

We thank the reviewer for their useful and insightful comments. Here we outline our responses in blue.

*The paper presents an extensive and well detailed model for NH3 emissions from chickens. As mentioned in the manuscript, most of the current global emission inventories for livestock are based on emission factors without any consideration for regional climate conditions and farming practices. The introduction of a few regional dependent parameters should greatly improve the spatial and temporal variations in the NH3 emissions which is of great value in for example in air quality modelling. The manuscript is well written, structured and easy to read.*

*Major comments*

*1. Hourly and Daily timescales; the authors describe in section 2.1 that their model operates at an hourly timescale for outdoors emissions while only at a daily level for in- door emissions. While variations in temperatures inside can reasonably be expected to be small, emissions will show some variations as a function of the inside temperature, which will lead to variations in the emissions to the outside. If possible add a sentence about the choice for two different timescales and the potential impact.*

We agree with the reviewer that variations in the indoor temperature of the houses will lead to variations in the emissions. The reason why the housing simulation was calculated at daily timescales is because the generalised relationship between indoor and natural temperature was derived from daily measurement data, and we want to keep the modelling structure consistent. Since the inside temperature is controlled, with variations typically smaller than diurnal variations of the outside temperature, we think simulating housing emissions with daily resolution can provide reasonable outcomes. In contrast, simulations of emissions from land spreading and backyard chickens were run with hourly timesteps in order to replicate the meteorological effects to capture diurnal variations. We propose to add a sentence in the methodology (Section 2.4.2 Global upscaling under Section 2.4 Global applications) to address this point.

*2. Section 3.1.3; line 17. I would argue that the model does an average job at capturing the overall level of emissions but that some of the major changes at the start of the measurement period seem to be over and underestimated by up to a factor 2 (figure 5 May-June and ~*

*September). Add some discussion on the main cause for the discrepancy between the modeled and measured emissions.*

We thank the reviewer for pointing out the discrepancy between modelled emissions and measurements in Fig 5. The model overestimated $NH_3$ emissions from early April to early July and then underestimated the emissions in September for House B. Following the suggestion of the reviewer, we propose to add a sentence noting this in the Results (Section 3.1.3 Resistance within chicken houses and site simulations). We know the discrepancies are mainly caused by the use of a fixed housing resistance, $R^*$. In reality, $R^*$ will vary with the environmental conditions within chicken houses. However, we consider it well-justified to use a constant value of $R^*$ in order to keep simple the overall fit of the dataset to the measured emissions, which also simplifies the global application. We agree that the value of $R^*$ and its variation across chicken house designs is a significant source of uncertainty in our results. For this reason, we have given attention to discussing the model uncertainty related to the $R^*$ value in the revised manuscript.

*3. Uncertainties; the various uncertainties within the model are discussed to great extent but what is missing is a final summary and overall estimate of uncertainty. If possible add a table summarizing the various errors and uncertainties, including an expected (back of the envelope) range of uncertainty for each individual error. Similarly add a summary/discussion on the total expected uncertainty, and a summary for the uncertainties in the spatial and temporal distributions (similar to the ranges, at a back of the envelope level).*

We thank the reviewer for this invaluable comment. We agree that there is substantial uncertainty in modelling $NH_3$ emission from livestock farming. Here, we focus on discussing the uncertainty related to model parameterizations. As stated by the reviewer, it is helpful to include a "back of the envelope" calculation of the overall uncertainty and uncertainties for individual components. The model parameters may influence the emissions interactively with non-linear consequences. We find that it is probably impossible to estimate the error based on mathematical approaches because the uncertainty distribution for many of the model terms is not well known. Instead, we conduct sensitivity analysis by simulating the effect of changes in parameters on $NH_3$ emissions. By doing this, we are able to indicate the ranges of uncertainty and also to highlight which parameters are most important and need to be further investigated. Based on prior test, we find that indoor resistance $R^*$, rearing density, manure pH, runoff coefficient and amount of N excreted are most important and examine these in the sensitivity

tests. In addition, the uncertainty arising from the parameterization of UA hydrolysis is represented by the differences between Fig. 8 and Fig. S9. Uncertainty related to human management and processes that are not included in the model are not quantitatively investigated here but has been discussed in the manuscript.

[revised manuscript text omitted]

Table R1.2 (manuscript Table 1) Excreted nitrogen from housed and backyard chicken, and estimated annual NH$_3$ emissions from each practice based on 2010

| Production system | Total excreted nitrogen (Tg N) | Practice | Total emission (Tg N) | Average $P_V$ (%) |
|---|---|---|---|---|
| Broiler and layer | 9.0 [±0.9] | Housing | 2.0 [±0.6] | 22 [±7] % |
| | | Land spreading | 2.7 [±0.5] | 39 [±7]* % |
| Backyard chicken | 2.2 [±0.2] | Left on land | 0.7 [±0.2] | 32 [±7] % |
| Total | 11.2 [±1.1] | | 5.5 [±1.2] | 49 [±11] % |

* Based on the excreted N remaining (i.e., 7.0 Tg N) after NH$_3$ volatilization from housing.

4. *Current inventories; that brings us to a comparison to current inventories which is as of yet missing in the manuscript. Most regional/country scale inventories, to some extent, do have emission totals for chicken housing/open-range chickens. How do the emissions reported in this manuscript compare to some of those emission inventories (for example, UK, Netherlands,*

*Denmark, US, German inventories. . .etc), and did the added complexity of the model improve the overall uncertainty in the emission totals?*

We thank the reviewer for this comment. We reply to this point together with *Comment 5*, please see our answer below.

*5. Similarly, add some discussion on the average Volatilization levels reported in this study compared to those in current literature.*

We thank the reviewer for these comments that comparing the results with existing inventories and literature. Here we can compare the results from the AMCLIM model to three other (model-based) studies/reports from Denmark, Netherlands and United Kingdom, respectively. The Danish IDA model (Albrektsen et al., 2017) and the UK NARSES model (Misselbrook et al., 2011) provided 2010 emission data, and the NEMA model (Velthof et al., 2012) from Netherlands estimate emissions in 2009 (see Table R1.3 below). It is important to clarify that all these studies show emissions from poultry rather than chicken. It has been clearly stated that the input used in the AMCLIM from the GLEAM model used here are chicken data, which excluded other poultry such as turkeys, ducks etc. Therefore, we can see that the excreted nitrogen from the GLEAM model (GLEAM FAO, 2018) is generally smaller than other individual studies.  For housing, the AMCLIM model shows similar estimates of $NH_3$ emissions to the other models. The housing emissions given by the AMCLIM model are smaller than the local models in Denmark and Netherlands, partly due to the smaller total excreted N from animals. However, the AMCLIM model suggests larger emissions from land spreading for Netherlands and the UK (spreading-derived emissions are not available from the IDA model), especially in Netherlands where the difference between the two estimates reaches 8 x. This is probably due to the different schemes or assumptions for land spreading practices, i.e. deep injection of manure, in different models. The $P_V$ rates, which indicate the fraction of nitrogen that is emitted as $NH_3$ are comparable from all models for the housing sector. The AMCLIM model suggests that the $P_V$ rates do not vary significantly between these countries because the indoor conditions are largely controlled and in similar climates, which leads to small variations in house environments. This table will be included in the revised manuscript. It should be noted that there is a lack of published experimental data on emissions from chicken in many climate (e.g. tropical climates), for which future measurements datasets would be useful to further test the model performance. We compare the model performance with experimental field studies in answer to reviewer 2 (see Figure R2.1 in reply 2).

Table R1.3 Estimates of $NH_3$ emissions from poultry/chicken farming by IDA for Denmark (Albrektsen et al., 2017) and by NARSES (Misselbrook et al., 2011) for the United Kingdom based on 2010, and by NEMA (Velthof et al., 2012) for Netherlands based on 2009*. Ranges given in the $P_V$-housing represents the geographical variations across the country.

| | Ammonia emission from Housing (Gg N yr$^{-1}$) | Ammonia emission from Spreading (Gg N yr$^{-1}$) | Total excreted N (Gg N yr$^{-1}$) | $P_V$-housing (%) |
|---|---|---|---|---|
| Denmark | 3.0 (IDA) | Not available | 11.3 (IDA) | 26.5 |
| | 1.7 (AMCLIM) | 2.4 (AMCLIM) | 7.9 (GLEAM) | 21.5 (20.4 – 22.9) |
| Netherlands | 11.4* (NEMA) | 1.8* (NEMA) | 62.9* (NEMA) | 18.1* |
| | 10.0 (AMCLIM) | 15.0 (AMCLIM) | 49.0 (GLEAM) | 20.4 (20.0 – 21.0) |
| United Kingdom | 15.0 (NARSES) | 14.7 (NARSES) | Not available | 17.8 |
| | 17.4 (AMCLIM) | 23.7 (AMCLIM) | 84.1 (GLEAM) | 20.7 (18.6 – 22.1) |

*Minor edits and remarks*

*a. Figure S1, is there any reasoning behind the choice of a third order polynomial?*

We use this third order polynomial equation to represent a generalised relationship between indoor and outdoor temperature because 1) it is roughly consistent with a simplified parameterization proposed by (Gyldenkærne et al., 2005) that the indoor temperature behaves in a "increase-stay-increase" pattern, 2) and it is applicable and convenient for computing. We will add this clarification to the revised manuscript in Section 3.1.1 Temperature of chicken houses.

*b. Page 6., line 9, add "of" between lack and knowledge.*

Corrected, thanks.

---

## Author Comment (AC2) · 30 Sep 2020

We thank the reviewer for their useful and insightful comments. Here we outline our responses in blue.

*General comments*

*The manuscript "A climate-dependent global model of ammonia emissions from chicken farming" from Jize Jiang et al., describes a model of ammonia volatilization from chicken farming: AMCLIM- Poultry.*

*The model is based on a simple approach were urea hydrolysis to ammonium and ammonia is implemented for emissions in buildings, in field applied with chicken manure and in farm backyards. A resistance approach is used and specific resistance parameterisation is used for buildings. A simple mass balance approach is used to treat manure water content.*

*The model is compared to measurements in a few US farms and applied to evaluate worldwide emissions from chicken farming, based on FAO statistics.*

*The issue is of great interest for the scientific community as ammonia emission is a key component of air quality prediction and environmental impacts and emissions from chicken farming is still not well developed. The presented study is based on the work of Elliot and Collins (1982) for hydrolysis and combined with a resistance approach. The application of the model at the global scale is of great interest, and especially the analysis of the humidity and temperature dependent NH3 emissions as well as the dataset constructed for that purpose.*

*This manuscript should be published provided some the authors answer some comments on the model design.*

*• Model: The model is key in this manuscript and it is both very simple but it accounts for the most important processes about the environmental conditions, which makes it effectively very useful. The presentation of the model may however be improved by first exposing clearly, right at the beginning, the hypothesis behind it, second condensing the description in the material and methods only, whereas it is now split between sections, and third, better explicating the model for manure spreading in the field.*

We thank the reviewer for these constructive and insightful comments; our reply is listed in detail below.

*o Regarding model hypothesis, I found several hypotheses that were not always explicit: i)there is no transfer resistance in the litter itself (eq. 7); ii) ammonium is considered the only form of TAN in the liquid phase (eq. 6); iii) the pH is considered not influenced by the UA hydrolysis; iv) NH4+ is consider to be completely free in the litter and soil and not to be bound to soil or litter particles; v) the system is considered to be litter only but no soil; vi) No exports are in the equations but the model is initialised at each house cleaning; vii) there no litter evaporation is considered in the houses, rather an equilibrium is considered.*

We thank the reviewer for pointing out that these hypotheses need to be explicitly described in the manuscript. We will update the manuscript to include briefly the following points in the methods section (according to the numbering used above by the reviewer):

i) There is no explicit term for transfer resistance in the litter that is simulated in the model. Instead, the housing resistance R* is considered to include an "integrated" resistance that consists of aerodynamic and boundary layer resistances and also the resistance of litter.

ii) In the model version used in the initially submitted manuscript, we considered that aqueous TAN is mainly in the form of $NH_4^+$. For the revised manuscript, we have now improved the model by including the dissociation constant for $NH_4^+$ ($K_{NH4}$) and generalise the Eq.6 as follows,

$$\Gamma = \frac{[NH_4^+]}{[H^+]} = \frac{[TAN]}{K_{NH_4^+} + [H^+]} = \frac{M_{TAN}}{V_{H_2O}(K_{NH_4^+} + [H^+])}$$

iii) We used a fixed pH of 8.5 rather than including a dynamical scheme for determining the pH. We appreciate that pH increases as UA hydrolyses, which causes larger instantaneous $NH_3$ emissions, similar to the effect simulated for urea by Móring et al. (2016). However, such an approach substantially complicates the model and involves substantial additional unknowns. For a practical model targeted for global upscaling, we therefore consider this simplification appropriate. We find that the changing the pH of the manure by ±1 causes the annual $NH_3$ emission to change by -15.9 % to 5.8 %. While the time course of instantaneous emissions changes, the uncertainty in the annual emission is smaller than the instantaneous effect, as this is constrained by the total amount of UA hydrolysed.

iv)    We simplified soil processes when simulating $NH_3$ volatilization from manure spreading. The volatilization of $NH_3$ is considered to be a much quicker process compared to the immobilization of $NH_4^+$ in the soil. In addition, the adsorption of TAN to soil is not simulated in this model because it requires detailed soil chemistry which is only achievable by using more detailed land models. This could be a future direction of study, also considering the effect of manure incorporation into the soil.

v)     We considered that manure or litter is the major substrate of TAN. This can be true because 1) there is no soil in chicken houses and 2) chicken excretion is relatively dry and with large fraction of solid materials compared to other livestock. The model thus cannot simulate interactions between manure and soil, after spreading. As mentioned above, this could be a potential future area of model development.

vi)    We do not include an export term in the mass balance equations. Instead, we set each pool to zero when there is an emptying event. The assumption is that when the houses are cleared out, this is complete, and all the cleared manure ends up being spread on local fields under current model resolution of $0.5 \times 0.5$ degree.

vii)   We do not simulate litter evaporation explicitly in houses because the model for housing simulation is run at daily time basis. The chicken excretion is relatively dry, and we assumed there is no extra water added to the system. It is a simplification that the manure has equilibrium moisture content after a day. The uncertainty has been discussed in the manuscript.

As requested by the reviewer, we will update the manuscript and clearly state the points above in the methods section.

*o Regarding the description of the model, it would be much easier to read if the whole model could be defined at once in the material and methods: factors affecting UA hydrolysis should be presented in the material and methods. Watch out that the TAN is sum of NH3l+ NH4 and you should justify NH4 >> NH3.*

We will update the manuscript to present the factors affecting UA hydrolysis in the Methods section. While we agree that $[NH_4^+] \gg [NH_3]$ we have now also updated Eq.6 (see point ii above) to better simulate the partition between $NH_3$ and $NH_4^+$.

*o Regarding the manure spreading, it is unclear how VH2O is calculated in this situation, and the description of run off is quite unclear.*

The $V_{H_2O}$ in outdoor simulations (manure spreading + backyard chicken) is calculated from the mass of water in the system, $M_{H_2O}$, from Eq. 14. The runoff is determined from a runoff coefficient multiplied by the amount of water that is available for runoff, which is determined by subtracting the water absorbed by the manure from the rainfall. We will update the manuscript to make this explicit.

- *UA hydrolysis fitting to RH and TA: Did you try fitting on vapour pressure pvap = RH/100\*psat(Ta) ? In addition, did you try fitting on both Ta and RH together?*

The RH and temperature dependence of UA hydrolysis are taken from Elliott and Collins (1984) and Riddick et al. (2017). Both studies used a combined influence, which is a product of individual factors as expressed by the Eq. 20. The impact of RH on UA hydrolysis is associated with the equilibrium moisture content, which depends on temperature and RH. We do not fit on multiple variables simultaneously. Instead, we decomposed the effects from each factor to normalise the UA hydrolysis rate. We appreciate that fitting UA hydrolysis to vapour pressure as well as vapour pressure deficit could be a future investigation.

- *Literature: I feel that some important papers may be lacking. In particular, on ammonia emissions data and models from land spreading manure or urea hydrolysis. The literature is much more abundant on dairy cow or pig manure, but I was wondering if and why it would not be possible to refer to these when building up the model for chicken manure. Some examples given here*

We thank the reviewer for listing these useful articles. We will discuss and include relevant papers. Sigurdarson et al. (2018) presented a comprehensive review for ammonia emissions from urea hydrolysis, which implicates important mitigation measures. McQuilling and Adams (2015) developed a model for estimating $NH_3$ emission from livestock in the United States. The paper is developed from McQuilling's PhD thesis that established an emission inventory for the US including poultry. We also use the paper of Miola et al. (2014) and literature cited therein to further evaluate our model performance for field application of poultry litter.

o *Ammonia Volatilization after Surface Application of Laying-Hen and Broiler-Chicken Manures. By: Miola, Ezequiel C. C.; Rochette, Philippe; Chantigny, Martin H.; et al. JOURNAL OF ENVIRONMENTAL QUALITY Volume: 43 Issue: 6 Pages: 1864-1872 Published: NOV-DEC 2014. Typos: please check thoroughly the text for typos.*

*o The molecular processes of urea hydrolysis in relation to ammonia emissions from agriculture By: Sigurdarson, Jens Jakob; Svane, Simon; Karring, Henrik. REVIEWS IN ENVIRONMENTAL SCIENCE AND BIO-TECHNOLOGY Volume: 17 Issue: 2 Pages: 241-258.*

*o Modeling and measurements of ammonia from poultry operations: Their emissions, transport, and deposition in the Chesapeake Bay By: Baker, Jordan; Battye, William H.; Robarge, Wayne; et al. SCIENCE OF THE TOTAL ENVIRONMENT Volume: 706 Article Number: 135290 Published: MAR 1 2020*

*o Semi-empirical process-based models for ammonia emissions from beef, swine, and poultry operations in the United States By: McQuilling, Alyssa M.; Adams, Peter J. ATMOSPHERIC ENVIRONMENT Volume: 120 Pages: 127-136 Published: NOV 2015*

*• Consider shortening the discussion. I found the discussion a bit long with a few redundancy and repetitions.*

We will update the discussion to make it more concise.

*• A comparison with existing emission factors would be very interesting*

We thank the reviewer for this insightful comment. We will add a comparison with existing emission factors. In particular, we take note of the review of experiments by Moila et al. (2014) and have addressed this further for inventories below.

*• Typos and English. I suggest double-checking the spelling and phrasing of the manuscript.*

We thank the reviewer for this considerate suggestion, and we will update the manuscript.

*Detailed comments*

*P2.L17-18: Could you be more specific on which parameters were tested?*

The effect of temperature and slurry dry matter content on $NH_3$ volatilization were based on the review of Sommer and Hutchings (2001). We will mention this in the revised manuscript.

*P3.Eqns (1-3): In these two equations, the export flux of excretion by removal during house cleaning is not considered. It would be clearer to add it. This would allow all Mexctretion, MUA and MTAN to get down to zero when the house is cleaned.*

Agree. We set the N pools to zero when the house is cleaned and will make this explicit.

*P4.L1: FTAN is not a conversion rate but a flux. Please consider revising.*

Agree. We will correct and update the manuscript. We change "$F_{TAN}$ is the conversion rate of UA to TAN" to "$F_{TAN}$ is the flux of TAN that is decomposed from UA hydrolysis".

*P4.L11: and eq. 4: it would be good to give expression of K here rather than in the results section.*

Agree. We will move this part to the method section.

*P4eq. 6 is not strictly speaking true since MTAN = MNH4+ + MNH3. Does this mean you consider MNH3 negligible compared to MNH4+? You could easily express MNH3 as a function of MNH4+ based on the dissociation constant and pH and then get a corrected expression for equation 6 that accounts for the pH.*

Agree. As answered previously, we have corrected the Eq.6 to include the dissociation constant for $NH_4^+$, which then allows both $NH_3$ and $NH_4^+$ to be included.

*P4L26-27: the justification of using the same approach for backyard and field may be more developed. Especially, how the interaction with the soil is treated.*

The same approach used for simulations of land spreading and backyard chicken refers to the broad resistance approach, which differs from the indoor resistance R* method. In this study, the interaction with the soil was not simulated, which is consistent with the GUANO model described by Riddick et al. (2017) which was validated for measured $NH_3$ emissions from seabird guano. The major difference between land spreading and backyard chicken is that we incorporated crop calendar dates to determining the timing of manure application for land spreading, whereas for backyard chicken excreta is deposited to land all year. Whereas ultimate immobilization, plant uptake or nitrification of TAN in the soil are not treated (since these are typically slower processes than $NH_3$ volatilization), these loss terms can be considered

implicitly as part of the uncertainty associated with depletion of deposited excreta by run-off. We will outline these points in the revised discussion, while further assessment of these interactions offers scope for future work

*P5L7-8: NH3 is removed but also fresh air dilutes NH3 in the building: both process occur.*

Agree. We will rephrase and update the manuscript.

*P5 eqns 8 and 9: From what I understand here, the litter (or excretions) has a humidity, which is in equilibrium with atmospheric humidity in the building (express by RH and T). This is similar to soil surface humidity that is in equilibrium with the atmosphere just above. Could-you explain the process behind equation 9?*

Equation 9 is based on the hygroscopicity of poultry litter and so accounts for the moisture absorbed by the litter as it reaches an equilibrium state, which is dependent on temperature and RH. The litter moisture content exerts a vapor pressure on the adjacent air, and the ratio of this moisture vapor pressure to the saturated vapor pressure of pure water in air at the temperature of the material is called the equilibrium relative humidity (Henderson, 1976). If the air RH is higher than the equilibrium relative humidity of the material, the material will increase in moisture content. Conversely, the material will decrease in moisture content if the air RH is lower than the equilibrium. We assume that the litter moisture content instantaneously maintains equilibrium with the housing environmental temperature and humidity, which we will clarify in the revised manuscript.

*P6L1: The pH should be influenced by UREA hydrolysis, isn't? Could you better justify the choice of fixing the pH?*

As answered previously (by iii), we do not include a dynamical scheme for determining the pH that can be influenced by the UA hydrolysis. We choose a fixed pH value of 8.5 to represent the system pH, which is a typical value of chicken excretion pH (Elliott and Collins, 1982). This is much more practicable for a global model than attempting to simulate explicitly the dynamic pH response of litter to UA hydrolysis, which depends on poorly known buffering capacity and may also vary between microsites (Móring et al., 2016). By carrying out sensitivity tests, we find that varying pH only leads to small change in total annual $NH_3$ emissions, where increasing pH leads to larger emissions over a shorter period, while reducing

pH because leads to slower but more sustained emissions. Increasing pH from 8.5 to 9.5 cause annual $NH_3$ emission to increase by 5.8 %, and a decrease of pH to 7.5 leads to a decline of emission by 15.9 %.

*P6L28: I suggest explicitly stating that Qxout has been neglected.*

Agree. We will explicitly state that Qxout has been neglected due to the negligible ambient concentration of $NH_3$ compared to indoor concentration. We will update the manuscript.

*P6 eq 12-13: fundamentally, this equation would also hold for water in buildings: hence, humidity in the building may depend on the rate of air renewal and the surface humidity. This would mean that pvapin = f(pvapout, Q, R\*, pvapsurface) but also that there would a removal flux for humidity also. Proportional to Q\*(pvapin-pvapout). Could you elaborate on that and justify better, why evaporation from building is neglected?*

As answered previously (by vii), we do not simulate litter evaporation in houses because the model for housing simulation is run on a daily time basis. The chicken excretion is relatively dry, and we assumed there is no extra water added to the system. It is a simplification that the manure has an equilibrium moisture content after a day. The uncertainty has been discussed in the manuscript.

*P7L3-4: I suggest defining clearly, what the "system" is: is it the litter only, or the litter plus a certain depth of soil?*

The system refers to the manure only, and soil processes are not simulated in the model. We will clarify the system definition in the manuscript.

*P7L8-9: Could you explain better why the water amount in the system could not be less than that in the excretion? Indeed, since evaporation occur, the water amount may become lower.*

As mentioned previously, we assume that the litter moisture content is in equilibrium with the environment. The model precludes a dynamic evaporation simulation for the litter. The litter tends to get drier if the humidity falls, and wetter if the humidity increases. The amount of water of the system should not be less than the equilibrium moisture content of the excretion. We will update the manuscript to clarify.

*P7, section 2.3: The field application is unclear and would need further details: 1) TAN in soil is known to be in equilibrium with clay, explain why this process is neglected. 2) The evaporation equations as well as the expressions of the resistances are not given and should be detailed, in the supplementary at least. 3) How is VH2O calculated in that situation?*

1) As noted above, the AMCLIM model does not include an interactive scheme for TAN and soil. We consider that chicken manure is mainly lying on the surface of crop lands because it is relatively dry and is not physically mixed with underlying soils. This means that the model as presented does not consider the potential benefit of immediate incorporation of poultry litter into soil. Meanwhile, simulating the interactions with soil would require a more detailed characterization of soil chemistry, which might only be achieved by employing a sophisticated land model. Therefore, we exclude soil processes that require more detailed information of soil properties, which is beyond the capability of this model. 2) Compared to the housing simulations that use equilibrium moisture content, for simulations of land spreading and backyard chicken, we used the evaporation data from ECWMF to determine the water pool. The resistances ($R_a$ and $R_b$) for $NH_3$ volatilization are calculated based on Seinfeld and Pandis (Seinfeld and Pandis, 2016). We will add a description of resistances in the supplementary materials. 3) As answered previously, the $V_{H_2O}$ in outdoor simulations (manure spreading + backyard chicken) is calculated from the mass of water in the system, $M_{H_2O}$, from Eq. 14. The runoff is determined from a runoff coefficient multiplied by the amount of precipitation water that is the rainfall subtracts the water absorbed by the manure. We will make this explicit in the revised manuscript.

*P8L28: but evaporation ay also occur in the building. Please comment.*

As answered previously, we assume that the litter moisture content is in equilibrium with the housing environment. We used the equilibrium moisture content to determine the water content of the litter.

*P8L30: "houses were empty in different months". Please rephrase as this is unclear what it means.*

The context is as follows: "12 simulations were run by assuming that chicken houses were emptied in different months for each simulation, i.e. from January to December, and the simulations started in corresponding month." To clarify our message, we will change this as

follows in the revised version: "To calculate the varying impacts of emptying the chicken houses at different times of the year, we ran 12 different year-long simulations: each starting from a different month, i.e. from January to December, and assuming the chicken house had just been emptied."

*P9eq 18: I suggest using the term Navailable instead of Nsoil_poultry. It is also unclear from the text, whether N_total includes manure and mineral nitrogen*

We change the $N_{Soil\_poultry}$ to $N_{available}$. $N_{total}$ includes nitrogen from manure fertilizer, of which nitrogen from chicken manure is only a small fraction considering the model grid resolution and the spatial distribution of other sources.

*P10L21-22: It is unclear when the building temperature is not used, what temperature is then used? Please clarify.*

A distinction needs to be made here between: i) the derivation of relationships between in-house and outdoor temperature for the model parametrization and ii) running of the AMCLIM model for global upscaling. The text here refers specifically to the former. In this case, the data for when broilers are <0.5 kg per bird are excluded from the parametrization because a) broilers smaller than this size do not contribute significantly to $NH_3$ emissions and b) houses are kept warmer than normal for the smallest chicks was compared with birds >0.5 kg. By excluding these data for small birds, a much better relationship can be found between indoor and outdoor temperatures (Fig. S1), which is also representative of the periods of significant $NH_3$ emissions. In running the AMCLIM model for global upscaling, the same relationship from Fig. S1 is applied for all weights of birds. This will tend to underestimate the temperature in houses for birds <0.5 kg, but as noted this will have negligible effect on total emissions, because these are dominated by periods when chicken are >0.5 kg weight. We will clarify this in updating the integrated description of the methods.

*P10-P11: section 3.1.2 should be in the material and methods section and not in the results as it is a model description to me.*

Agree. We will move this to the method section.

*P11 eq 23: To me it would be more logical if urea hydrolysis would be dependent on the excretion humidity %me rather than RH. However, the two are linked. Could you comment on that?*

As noted above, the housing model is run on a daily time-step, since this is the time-scale for which we have measured emission data for verification. This means that we need to identify a representative litter humidity for daily periods for use with the parametrized relationship between litter humidity and hydrolysis rate. Bird excreta is actually liquid, but the water will be dispersed in a litter-based system throughout the litter. If it is envisaged that fresh excreta reaches equilibrium with the surrounding litter within an hour or a few hours, then this means that for a daily simulation it is more representative to use the litter humidity in equilibrium with daily humidity data. We will add a comment to this effect in the methods.

*P11-L16-17: "emissions were due to unavailable measurements": this sounds weird: could you rephrase?*

For the revised manuscript we propose to change "Gaps occurred in measured $NH_3$ concentration and emissions were due to unavailable measurements, while the model was kept running." into "Gaps shown in measured concentrations and emissions of $NH_3$ represent unavailable measurements, while the model was kept running during gaps to produce continuous output."

*P12 section 3.3: the model for manure spreading was not tested at all, while the model for housing was tested. Would there be any dataset to demonstrate the quality of the model for outdoor application? Alternatively, would there be any paper to refer to on that?*

We will make it clear that, from an experimental perspective, the AMCLIM model builds on the approach of the GUANO model, which has been tested in a wide range of outdoor climatic conditions (Riddick et al., 2018). In addition, we propose to include a brief comparison with the studies summarized by Miola et al. (2014), based on comparison of the $P_V$ values (i.e. % of TNA of Miola et al., % of Total N applied).

To address this, we ran a set of simple site experiments for land spreading to quantify the $NH_3$ volatilization rates ($P_V$) under different environmental conditions. We set the application rate to 100 kg N ha$^{-1}$ (equivalent to 10 g N m$^{-2}$), which is comparable to the value used in Rodhe

and Karlsson (2002) (110 kg N ha$^{-1}$), Sharpe et al. (2004) (109 kg N ha$^{-1}$, 99 kg N ha$^{-1}$, 133 kg N ha$^{-1}$) and Marshall et al. (1998) (70 kg N ha$^{-1}$). The model is driven by the mean daily air temperature given from the previous studies, while the diurnal variations of temperature and other meteorological factors such RH and precipitation are not available from these publications. The ground temperature is assumed to be 2 ° C higher than the air temperature, where ground temperature is not available from the published experiment. The sum of aerodynamic and boundary layer resistances is assumed to be 100 s m$^{-1}$ as it cannot be calculated due to the lack of environmental inputs provided by the authors. The wash-off pathways of the model were shut down due to the unknown rainfall information, so the simulations are representative of rain free experimental conditions. We initialized the model simulation using a 7-day period prior to application of chicken litter, to allow initialisation for each nitrogen pools. The model was then run for 21 days to determine the NH$_3$ volatilization. We compare the modelling results with reported measurements from five experimental studies (Lau et al., 2008; Marshall et al., 1998; Miola et al., 2014; Rodhe and Karlsson, 2002; Sharpe et al., 2004), as shown in Fig. R2.1. We focus on experimental data for chicken that are broilers or layers (rather than other poultry, e.g. turkey) and data for "young" litter which was stored for a short period before application, normally less than a week or 10 days. There are three groups of comparisons that represent different simulation and measurement duration.

As shown in Fig. R2.1, the simulated volatilization rate of NH$_3$ increases as temperature increases, because of the faster UA hydrolysis rate in hotter conditions. The shaded areas illustrate ranges of P$_V$ from simulations that use different RH values ranging from 20 to 100 %, while the solid lines represent the mean P$_V$ rate for the range of RH values for each simulation period (7, 14, 21 days).

Compared with the experimental studies shown in Fig. R2.1, the model application underestimates NH$_3$ volatilization for the 21 days simulation and overestimates for the 14 days simulation. However, it is evident that these experimental studies also show large variations, which we expect is especially due meteorological variation within and between the experimental studies, such as rainfall or windy conditions. For example, at a mean temperature of around 26 °C Sharpe et al. (2004) reported P$_V$ of 23 % and 5 %, respectively. The latter value was caused by a rain event taking place two days after application, explaining why the latter point appears low on Fig. R2.1 where the simulations are based on rain free conditions.

Overall, the model provides $P_V$ rates that falls within the range between 0.5 x to 2 x compared to the measurements. It should be noted that this is a very simple model experiment as several features of the AMCLIM-Poultry are not available because the published experimental studies do not fully describe environmental conditions.

[Figure]

Figure R2.1 Simulated fraction of total applied nitrogen that is loss as $NH_3$-N ($P_V$, %) as a function of air temperature (°C) by the AMCLIM-Poultry for simulating periods of 7, 14 and 21 days, and comparison with experimental studies that measured $NH_3$-N loss for 7, 14 and 21 days. Simulations conducted for rain-free conditions, where shaded areas indicate the range for simulations from 20 % to 100% relative humidity. The figure of 5 % volatilization at 27 °C by Sharpe et al. (2004) was associated with high precipitation.

*P14- L6-13: it is actually unclear in the previous part if the papers how RH and Ta are modelled in houses.*

We used the outdoor RH to represent the indoor RH for the housing simulations because the indoor and outdoor RH were found to be comparable from the USEPA AFO's dataset. The indoor temperature was determined by using generalised relationships shown in Fig. S1 based on AFO data. We will make this clearer in the revised methods section.

*P14-L14-20: I would suggest adding a table with durations, temperatures and may be RH conditions for the different chicken houses managements discussed*

The environmental variables of the houses including temperature and RH vary with time. We have shown the variations in Figs. 4 and 5.

*P14-15 section 4.1: it is a bit confusing here to understand how the RH-dependency of urea hydrolysis is used in outdoor conditions. Please detail.*

Section 4.1 is the discussion of parameterization of housing simulation instead of outdoor simulations. We simulated $NH_3$ emissions from chicken housing by using both the RH dependency of UA hydrolysis from Elliott and Collins (1982) and that is derived from USEPA AFO's dataset. The results are shown in Fig. 8 and Fig. S9, respectively. The RH dependency of UA hydrolysis used for outdoor simulations is from Elliott and Collins (1982), which has been previously tested and found to provide robust estimates from the GUANO model (Riddick et al., 2017). We will clarify the text accordingly.

*P16L1-10: the whole paragraph except last sentence is quite unclear. Please rephrase. In the last sentence, it may not be true that sensitivity is negligible though, since R\* may be very variable among situations.*

We will rephrase this paragraph. We have now carried out a set of sensitivity tests for global simulations that detail how $NH_3$ emissions vary with several uncertain parameters (Table R2.1). We find that varying indoor resistance values, R\* by a factor of 2 causes $NH_3$ emissions to change by approximately 30 %: 2x higher R\* leads to $NH_3$ emission decrease by approximately 30 %, and 2x lower R\* leads to 27 % higher emissions, which is similar to the result of sensitivity test at the site scale.

*P16-L27-33: Could we not say that for very large RH, since UA hydrolysis is so effective, there is a limiting effect due to the non-availability of total nitrogen in the system after a certain time?*

We agree that this could happen in principle, but suggest that this cannot explain the results of our steady-state model run as summarized in Fig. 7b. Firstly, if total N were limiting, then this would mean that the value of $P_V$ would not increase further above a certain threshold. However, we see that the value of $P_V$ actually *decreases* above 80% RH, pointing to the need for a different explanation. As we have noted, with excess water available, there is a dilution effect on TAN concentration, which can explain this feature. Secondly, we would expect that total N would become limiting once all available UA is hydrolysed (equivalent to 60% volatilization rate of total excreted N). However, we do not find this threshold to be exceeded. Therefore, we consider the dilution effect to be the likely cause of this decrease in $P_V$ above 80% RH.

*P17L8-9: Difficult to understand. Please rephrase this sentence*

For the revised manuscript, we propose to change "Considering the variations in $P_V$, there is most estimated variation in NH3 volatilization of manure spreading and backyard." into "Considering the $P_V$, the most significant spatial variations relate to emissions from manure spreading and backyard chicken, with less spatial variation in $P_V$ for housed birds"

*P18L26: It is unclear why initial water in excretion is not accounted for. Please rephrase.*

We explain the reason in P18L24-25, "The model is not able to simulate the evaporation from the litter in the chicken house. Therefore, the litter moisture is assumed to be at equilibrium". As answered previously (reply to comment on *P11 Eq.23*), chicken excretion is relatively dry compared with other livestock excreta, so we assumed it takes a much shorter time for chicken litter to reach equilibrium moisture content than the modelling timestep (1 day), allowing use of the equilibrium value.

*P18-last paragraph: this section would need sensitivity tests to better demonstrate that R\* does not represent a great uncertainty.*

As answered previously, by carrying out sensitivity tests (Table R2.1), we find that 2x higher R\* leads to annual NH3 emission decrease by approximately 30 %, and 2x lower R\* leads to 27 % higher emissions. The annual effect is smaller than the instantaneous response because lower emissions tend to be more sustained and vice versa.

*P19 section 4.3.2: In this section a sensitivity to pH would be interesting to show to illustrate the possible effect of changing the manure pH by +- 1 point.*

We carry out a set of sensitivity tests (Table R2.1). We find that increasing pH from 8.5 to 9.5 causes annual $NH_3$ emission to increase by 5.8 %, while a decrease of pH to 7.5 leads to a decline of emission by 15.9 % (as described above). As with R*, the sensitivity to pH is smaller for annual emissions as compared with instantaneous emission. More detailed discussion can be seen in the reply to Reviewer 1.

Table R2.1 Sensitivity test for model parameters for global application of the model.

| Parameter | Value tested | Value change | $\Delta NH_3$ emission % | |
|---|---|---|---|---|
| [a, b] Indoor resistance, R* | 16700 s m$^{-1}$ (base) | 1 x | 0.0 % | |
| | 8350 s m$^{-1}$ | 0.5 x | [a] 27.1 % | [a, b] 8.5 % |
| | 33400 s m$^{-1}$ | 2 x | [a] -30.6 % | [a, b] -6.4 % |
| [a, b, c] Manure pH ($H^+$) | 8.5 (base) | 1 x | 0.0 % | |
| | 7.5 | 0.1 x | -15.9 % | |
| | 9.5 | 10 x | 5.8 % | |
| [b, c] Runoff coefficient, $R_{runoff}$ | 1 % mm$^{-1}$ (base) | 1 x | 0.0 % | |
| | 0.5 % mm$^{-1}$ | 0.5 x | 16.5 % | |
| | 2 % mm$^{-1}$ | 2 x | -11.8 % | |
| [a, b, c] Excreted nitrogen | 11.2 Tg N year$^{-1}$ (base) | 1 x | 0.0 % | |
| | 10.1 Tg N year$^{-1}$ | 0.9 x | -12.3 % | |
| | 12.3 Tg N year$^{-1}$ | 1.1 x | 12.6 % | |

[a] Parameters affect $NH_3$ emissions from housing. [b] Parameters affect $NH_3$ emissions from land spreading of chicken manure. [c] Parameters affect $NH_3$ emissions from backyard chicken.

*FIGURES AND TABLES*

*Fig 1: explain meaning of arrows*

The arrows in Fig. 1 represent the nitrogen flows from chicken farming. We will update the figure caption of Fig. 1.

*Fig 2: I would suggest adding flows in and out of the farm. In addition, an arrow for dilution through ventilation pointing towards INDOOR NH3 LEVELS may be considered. Watch out that the volatilisation flux is bi-directional. An arrow downwards should be shown.*

Figure 2 shows critical processes of $NH_3$ emissions from chicken houses, which originates from chicken excretion. As we have not simulated other flows of N into our model out of the farm, we consider it better not to include such arrows. Process 1 represents the input of model that the nitrogen is in the form of UA from poultry excretion, and process 6 shows that the $NH_3$ emission is released from the houses to the outside atmosphere through ventilation (a flow out). The indoor $NH_3$ levels were simply calculated by dividing the $NH_3$ left in the house by the volume of the house. It may be noted that the arrow for process 6 is already connected to process 5.

Yes: we appreciate $NH_3$ fluxes can, in general, be both bi-directional, i.e. emission, or the reverse, deposition, and are dependent on the $NH_3$ concentrations in the surface source material and the overlying atmosphere. To reflect this point, we have referred at Eq. 7 to the study of Sutton et al. (2013) which considers this in detail. That paper also distinguishes between sources which are bi-directional (land surfaces) versus sources which are in effect only ever unidirectional (animal houses). For the situations in this study, i.e. $NH_3$ fluxes from N-rich animal excreta, we considered that chicken excretion is a strong source of $NH_3$ emissions from the surface, so we simplified the model to a uni-directional scheme. (We can envisage no practical case where outdoor atmospheric $NH_3$ concentrations would be larger than at the surface of chicken excreta). In order to be consistent with the model description, we do not include a downwards arrow in this situation.

*Fig 3: It is unclear how the UA factors were calculated. 3a: could you give a hint on the significance of the difference between the two curves?*

Figure 3a shows the relationship between the T factor of Elliot and Collins (1982) (red line) and that derived from the AFO experimental data (Section 2.2.1). (blue line). The blue line represents the least squares best-fit to the AFO data using a polynomial function of the form used by Elliot and Collins. It is possible to test whether the line of Elliot and Collins is significantly different from the data, by considering whether the mean difference (from the red line to points is significantly different from zero. For n=21, the mean difference in factor T between the red line and the data is 0.037 +/- 0.011 (standard error) which is significantly

different to zero with P>99% confidence. The value of Elliot and Collins is therefore significantly different from the AFO dataset.

*Fig 4d and 5d: I would suggest showing also on the same graph the ammonia concentration at z0 (the compensation point) as it would give ground to better understand the NH3 emissions dynamics.*

We will update the figures to include $NH_3$ concentration at $z_0$.

*Fig7: please explicit the fact that the curves are evaluated for yearly datasets. I suggest showing also total UAN remaining before cleaning to show any N-limiting effect on Pv. I also suggest rephrasing: 'NH3 volatilization rate Pv(%) for 4 different RH and Ta regimes....'*

Agree. We will update Fig .7. We change "Curves that represent 4 different regimes from Fig. 6." into "Curves that represent $NH_3$ volatilization rate $P_V$ (%) for 4 different temperature and RH regimes based on annual steady-state simulations (see Fig. 6)."

*Table 1: I would suggest adding percentage of N loss for each production system. In addition, you may consider getting rid of unneeded precision in emission numbers.*

Agree. We will update manuscript Table 1.

Table R2.2 (manuscript Table 1) Excreted nitrogen from housed and backyard chicken, and estimated $NH_3$ emissions from each practice (global estimates for 2010). Uncertainty indicate the combined uncertainty ranges based on model sensitivity tests (Table R2.1).

| Production system | Total excreted nitrogen (Tg N) | Practice | Total emission (Tg N) | Average $P_V$ (%) |
|---|---|---|---|---|
| Broiler and layer | 9.0 [±0.9] | Housing | 2.0 [±0.6] | 22 [±7] % |
| | | Land spreading | 2.7 [±0.5] | 39 [±7]* % |

| | | | | |
|---|---|---|---|---|
| Backyard chicken | 2.2 [±0.2] | Left on land | 0.7 [±0.2] | 32 [±7] % |
| Total | 11.2 [±1.1] | | 5.5 [±1.2] | 49 [±11] % |

\* Based on the excreted N remaining (i.e., 7.0 Tg N) after $NH_3$ volatilization from housing.

References

Elliott, H. A. and Collins, N. E.: Factors Affecting Ammonia Release in Broiler Houses, Trans. ASAE, 25(2), 0413–0418, doi:10.13031/2013.33545, 1982.

Henderson, S.: Agricultural Process Engineering, Springer US., 1976.

Lau, A. K., Bittman, S. and Hunt, D. E.: Development of ammonia emission factors for the land application of poultry manure in the Lower Fraser Valley of British Columbia, Can. Biosyst. Eng. / Le Genie des Biosyst. au Canada, 50, 47–55, 2008.

Marshall, S. B., Wood, C. W., Braun, L. C., Cabrera, M. L., Mullen, M. D. and Guertal, E. A.: Ammonia Volatilization from Tall Fescue Pastures Fertilized with Broiler Litter, J. Environ. Qual., 27(5), 1125–1129, doi:10.2134/jeq1998.00472425002700050018x, 1998.

McQuilling, A. M. and Adams, P. J.: Semi-empirical process-based models for ammonia emissions from beef, swine, and poultry operations in the United States, Atmos. Environ., 120, 127–136, doi:10.1016/j.atmosenv.2015.08.084, 2015.

Miola, E. C. C., Rochette, P., Chantigny, M. H., Angers, D. A., Aita, C., Gasser, M.-O., Pelster, D. E. and Bertrand, N.: Ammonia Volatilization after Surface Application of Laying-Hen and Broiler-Chicken Manures, J. Environ. Qual., 43(6), 1864–1872, doi:10.2134/jeq2014.05.0237, 2014.

Móring, A., Vieno, M., Doherty, R. M., Laubach, J., Taghizadeh-Toosi, A. and Sutton, M. A.: A process-based model for ammonia emission from urine patches, GAG (Generation of Ammonia from Grazing): description and sensitivity analysis, Biogeosciences, 13(6), 1837–1861, doi:10.5194/bg-13-1837-2016, 2016.

Riddick, S. N., Blackall, T. D., Dragosits, U., Tang, Y. S., Moring, A., Daunt, F., Wanless, S., Hamer, K. C. and Sutton, M. A.: High temporal resolution modelling of environmentally-dependent seabird ammonia emissions: Description and testing of the GUANO model, Atmos. Environ., 161, 48–60, doi:10.1016/j.atmosenv.2017.04.020, 2017.

Riddick, S. N., Dragosits, U., Blackall, T. D., Tomlinson, S. J., Daunt, F., Wanless, S., Hallsworth, S., Braban, C. F., Tang, Y. S. and Sutton, M. A.: Global assessment of the effect of climate change on ammonia emissions from seabirds, Atmos. Environ., 184, 212–223, doi:10.1016/j.atmosenv.2018.04.038, 2018.

Rodhe, L. and Karlsson, S.: Ammonia Emissions from Broiler Manure Influence of Storage and Spreading Method Lena, Biosyst. Eng., 82(4), 455–462, doi:10.1006/bioe.2002.0081,

2002.

Sharpe, R. R., Schomberg, H. H., Harper, L. A., Endale, D. M., Jenkins, M. B. and Franzluebbers, A. J.: Ammonia Volatilization from Surface-Applied Poultry Litter under Conservation Tillage Management Practices, J. Environ. Qual., 33(4), 1183, doi:10.2134/jeq2004.1183, 2004.

Sigurdarson, J. J., Svane, S. and Karring, H.: The molecular processes of urea hydrolysis in relation to ammonia emissions from agriculture, Rev. Environ. Sci. Bio/Technology, 17(2), 241–258, doi:10.1007/s11157-018-9466-1, 2018.

Sutton, M. A., Reis, S., Riddick, S. N., Dragosits, U., Nemitz, E., Theobald, M. R., Tang, Y. S., Braban, C. F., Vieno, M., Dore, A. J., Mitchell, R. F., Wanless, S., Daunt, F., Fowler, D., Blackall, T. D., Milford, C., Flechard, C. R., Loubet, B., Massad, R., Cellier, P., Personne, E., Coheur, P. F., Clarisse, L., Van Damme, M., Ngadi, Y., Clerbaux, C., Skjøth, C. A., Geels, C., Hertel, O., Wichink Kruit, R. J., Pinder, R. W., Bash, J. O., Walker, J. T., Simpson, D., Horváth, L., Misselbrook, T. H., Bleeker, A., Dentener, F. and de Vries, W.: Towards a climate-dependent paradigm of ammonia emission and deposition, Philos. Trans. R. Soc. B Biol. Sci., 368(1621), 20130166, doi:10.1098/rstb.2013.0166, 2013.